



# Reconstruction of global gridded monthly sectoral water withdrawals for 1971-2010 and analysis of their spatiotemporal patterns

Zhongwei Huang[1, 2, 5], Mohamad Hejazi[2,3], Xinya Li[4], Qiuhong Tang[1, 5], Guoyong Leng[2], Yaling Liu[2],
Petra Döll[6,7], Stephanie Eisner[8], Dieter Gerten[9,10], Naota Hanasaki[11], Yoshihide Wada[12]

[1]Key Laboratory of Water Cycle and Related Land Surface Processes, Institute of Geographical Sciences and Natural Resources Research, Chinese Academy of Sciences, Beijing, China

[2]Joint Global Change Research Institute, Pacific Northwest National Laboratory, College Park, MD, USA

[3]Earth System Science Interdisciplinary Center, University of Maryland, College Park, MD, 20740, USA

[4]Pacific Northwest National Laboratory, Richland, WA, USA

[5]University of Chinese Academy of Sciences, Beijing, China

[6]Institute of Physical Geography, Goethe University Frankfurt, Frankfurt am Main, Germany

[7]Senckenberg Biodiversity and Climate Research Centre (BiK-F), Frankfurt am Main, Germany

[8]Center for Environmental Systems Research, University of Kassel, Kassel, Germany

[9]Research Domain of Earth System Analysis, Potsdam Institute for Climate Impact Research (PIK), Potsdam, Germany

[10] Geography Dept., Humboldt-Universität zu Berlin, 10099 Berlin

[11]Center for Global Environmental Research, National Institute for Environmental Studies, Tsukuba, Japan

[12]International Institute for Applied Systems Analysis (IIASA), Schlossplatz 1, A-2361 Laxenburg, Austria

*Correspondence to*: Mohamad Hejazi (Mohamad.Hejazi@pnnl.gov)

**Abstract**

Human water withdrawal has increasingly altered the global water cycle in past decades, yet our understanding of its driving forces and patterns is limited. Reported historical estimates of sectoral water withdrawals are often sparse and incomplete, mainly restricted to water withdrawal estimates available at annual and country scale, due to a lack of observations at local and seasonal time scales. In this study, through collecting and consolidating various sources of reported data and developing spatial and temporal statistical downscaling algorithms, we reconstruct a global monthly gridded (0.5 degree) sectoral water withdrawal dataset for the period 1971–2010, which distinguishes six water use sectors, i.e. irrigation, domestic, electricity generation (cooling of thermal power plants), livestock, mining, and manufacturing. Based on the reconstructed dataset, the spatial and temporal patterns of historical water withdrawal are analyzed. Results show that global total water withdrawal has increased significantly during 1971-2010, mainly driven by the increase of irrigation water withdrawal. Regions with high water withdrawal are those densely populated or with large irrigated cropland production, e.g., the United States (US), eastern China, India, and Europe. Seasonally, irrigation water withdrawal in summer for the major crops contributes a large percentage of annual total irrigation water withdrawal in mid and high-latitude regions, and the dominant season of irrigation water withdrawal is also different across regions. Domestic water withdrawal is mostly characterized by a summer peak, while water



withdrawal for electricity generation has a winter peak in high-latitude regions and a summer peak in low-latitude regions. Despite the overall increasing trend, irrigation in the western US and domestic water withdrawal in western Europe exhibit a decreasing trend. Our results highlight the distinct spatial pattern of human water use by sectors at the seasonal and annual scales. The reconstructed gridded water withdrawal dataset is open-access, and can be used for examining issues related to water withdrawals at fine spatial, temporal and sectoral scales.

## 1. Introduction

With the rapid growth in population, income, and demand for energy, feed, and food, global freshwater withdrawal increased from ~2500 km$^3$ yr$^{-1}$ in 1970 to ~4000 km$^3$ yr$^{-1}$ in 2010 (Shiklomanov, 2000; Döll et al., 2009; Wada and Bierkens, 2014). Such large-scale human water withdrawals have significant impacts on both the water cycle, the associated ecosystems, and society. For example, irrigation has redistributed surface water and groundwater resources, and perturbed terrestrial hydrology via changes in evapotranspiration and streamflow (White et al., 1972; Stohlgren et al., 1998; Haddeland et al., 2006; Tang et al., 2008; Kustu et al., 2011; Wang and Hejazi, 2011; Döll et al., 2012; Taylor et al., 2013; Döll et al., 2014), which has in turn altered surface air temperature and precipitation at regional and global scale (Adams et al., 1990; Boucher et al., 2004; Kueppers et al., 2007; Lobell et al., 2009; DeAngelis et al., 2010). Rost et al. (2008) stated that irrigation increased global evapotranspiration by ~2% and decreased river discharge by 0.5% during 1971-2000, while Müller Schmied et al. (2014) computed an increase of global evapotranspiration due to human water use (with approx. 90% being due to irrigation) of about 1.3% and a decrease of river discharge of about 1.8 %. Furthermore, increasing human water withdrawals limit further economic development, particularly in arid or semi-arid regions, e.g., northern China, India, Middle East (Rodell et al., 2009; Wada et al., 2011; Taylor et al., 2013; Yin et al., 2017). Although characterizing the impact of human water use on the hydrological cycle would entail a comprehensive assessment of the water lifecycle from source (surface vs groundwater), to end use sectors (irrigation, industrial, domestic), to changes to its quality (waste water), to its eventual return to the environment (return flow) or consumption (consumptive use) (Wada et al., 2014), we focus in this study on water withdrawal.

During the past years, many global hydrological models (GHMs), land surface models (LSMs) and integrated assessment models (IAMs) have incorporated water management modules to assess global water withdrawal by sectors (Döll and Siebert, 2002; Tang et al., 2007; Hanasaki et al., 2008b; Rost et al., 2008; Wada et al., 2011; Pokhrel et al., 2012; Flörke et al., 2013; Hejazi et al., 2014). However, large discrepancies exist among different modeling studies with respect to the magnitudes of water withdrawals, due to differences in model structure, input parameters, climate forcing, and assumptions to supplement the data deficiencies (Wada et al., 2016). Therefore, cross-comparison of estimated water withdrawal from large-scale models is critical for quantifying the impacts of human water withdrawal, which was hampered so far due to a lack of water withdrawal benchmark at fine spatial and temporal scales (Barnett et al., 2005; Wada et al., 2011; Voisin et al., 2013; Hejazi et al., 2015; Leng et al., 2016).



Historical water withdrawal records by sectors are reported by many agencies or organizations. Shiklomanov and Rodda (2003) published a global water resources assessment (including water withdrawal and consumption data) for 26 regions according to literature review and statistical surveys. Additionally, estimated water use by sectors (irrigation, livestock, domestic, industry, and hydroelectric power) at state and county level in the United States has been reported by the US Geological Survey (USGS) every 5 years since 1950, and 1985, respectively. Similar historical water use reports are also published by the Ministry of Water Resources of China, the Statistisches Bundesamt of Germany, the Ministry of Land Infrastructure and Transportation in Japan, and the Water Security Agency of Canada. Another global water use inventory, AQUASTAT, which has been developed by the Food and Agriculture Organization (FAO), provides historical water withdrawals in particular sectors (agriculture, irrigation, domestic, and industry) every 5-year at country level. Unfortunately, these historical records in some regions or water use sectors are often incomplete or missing. Recently, Liu et al. (2016) developed a country scale water withdrawal dataset by sector at 5-year interval for 1973-2012 by filling the missing values in FAO AQUASTAT dataset. Furthermore, most existing water withdrawal inventories have been published at annual scale or 5-year interval for a particular region, which ignores the seasonal and spatial variations (aside from the irrigation estimates by models). The coarseness in data granularity may cause inadequate understanding for finer scale water use and hold back water management policy development.

Thus, establishing a comprehensive and consistent global dataset of historical water withdrawal time series, capturing both the seasonality and spatial variations, is important for multiple reasons. First, the reconstructed global historical gridded water withdrawal dataset can be used for cross–comparison of water withdrawal estimates of GHMs and also to supplement the water withdrawal estimates in LSMs due to lack of domestic and industrial water withdrawal simulation in most LSMs. Furthermore, such a dataset is important for investigating water use related issues and patterns at high spatial, temporal and sectoral resolutions, which is critical for developing sound water management strategies. The overarching goal of this study was to generate such a historical global monthly gridded water withdrawal data (0.5x0.5 degrees) for the period 1971-2010, distinguishing six water use sectors (irrigation, domestic, electricity generation, livestock, mining, and manufacturing).

The dataset constitutes the first reconstructed global water withdrawal data product at sub-annual and sub-national/gridded resolution that is derived from different models and data sources; it was generated by spatially and temporally downscaling country-scale estimates of sectoral water withdrawals from FAO AQUASTAT (and state-scale estimates of USGS for the US). In addition, the industrial sector was disaggregated into manufacturing, mining and cooling of thermal power plants. Downscaling was performed using the output of various models and new modeling approaches. This study adopts the spatial and temporal downscaling methodologies for water withdrawal in previous studies (Wada et al., 2011; Voisin et al., 2013; Hejazi et al., 2014; Wada and Bierkens, 2014), and further validates the temporal downscaling for water withdrawal domestic and electricity generation globally. Thus, with the application of the spatial and temporal downscaling methodologies, a reconstruction of global monthly gridded water withdrawal dataset for the period 1971-2010 is generated based on multiple reported data sources. Then the spatial and temporal patterns of global water withdrawal by sectors as provided by the newly



developed dataset are analyzed. In this paper, data and methods are described in section 2. Section 3 presents the spatiotemporal patterns of water withdrawal by sectors based on the newly developed dataset, and section 4 discusses the uncertainty and limitation of our work. Conclusions are presented in section 5.

## 2 Data and Methodology

### 2.1 Data

Water withdrawal in US is obtained from the USGS (http://water.usgs.gov/watuse/) at the state level for every 5 years since 1950, and by sector (irrigation, livestock, domestic, thermoelectric power, mining and manufacturing). In addition, FAO AQUASTAT provides water withdrawal data for agriculture, irrigation, domestic and industrial per 5-year interval for 200 countries (http://www.fao.org/nr/water/aquastat/data/query/), and the missing values were filled by Liu et al. (2016) using several techniques such as inverse weighting, linear interpolation, and proxies (e.g. irrigated land area, industrial value added, and population). Water withdrawal for electricity generation, mining and manufacturing are retrieved from the industrial sector in FAO AQUASTAT in combination with the sectoral water withdrawal simulation of the Global Change Assessment Model (GCAM). Here, water withdrawal datasets from USGS and FAO AQUASTA, which are used to reconstruct the global gridded monthly water withdrawal dataset, are applied in the US and in the rest of world, respectively.

The data sets used for spatial and temporal downscaling of sectoral water withdrawal are listed in Table 1. Global population density maps, which are applied for spatial downscaling of domestic, electricity generation, mining and manufacturing sectors, were obtained from the History Database of the Global Environment (HYDE) during 1970-1980 and Gridded Population of the World (GPW) during 1985-2010 in Socioeconomic Data and Application Center (SEDAC). Global livestock densities maps for 6 species (i.e. cattle, buffalo, goat, sheep, pig and poultry) for the year 2005 were collected from the FAO's Animal Production and Health Division. The gridded daily air temperature data from WATCH Forcing Data methodology applied to ERA Interim reanalysis data (WFDEI) from 1971 to 2010 is used for temporal downscaling of electricity and domestic water withdrawal from annual to monthly (Weedon et al., 2014). Other sources of air temperature data, from WATCH (Weedon et al., 2010), Princeton (Sheffield et al., 2006) and GSWP3 (Compo et al., 2011), are also adopted to examine the uncertainty of different climate forcing on simulated global monthly water withdrawal for electricity and domestic sectors. In addition, four global gridded monthly irrigation water withdrawal simulations for the period 1971-2010, which are obtained from the Inter-Sectoral Impact Model Inter-comparison Project (ISI-MIP) (Warszawski et al., 2014), are utilized for the reconstruction of irrigation water withdrawal. The four products were generated by 4 GHMs, i.e. WaterGAP (Döll and Siebert, 2002; Alcamo et al., 2003; Döll et al., 2009; Müller Schmied et al., 2014), LPJmL (Rost et al., 2008), H08 (Hanasaki et al., 2008a, b), and PCR-GLOBWB (Van Beek et al., 2011; Wada et al., 2011; Wada et al., 2014), and they are all forced by WFDEI climate data.



To investigate the uncertainty derived from forcing data, we also use other three simulated irrigation water withdrawal by WaterGAP forced by three datasets (i.e. Princeton, GSWP3 and WATCH).

## 2. 2 Methodology

Water withdrawal datasets from FAO AQUASTA and USGS need to be spatially downscaled from country (or state) level to grid scale, and temporally downscaled from 5-year interval to monthly scale. As for irrigation sector, correction factors are used to scale the irrigation water withdrawal estimates by GHMs according to reported data. For the other sectors, the spatial and temporal downscaling is applied to FAO AQUASTA and USGS estimates independently to get the monthly gridded dataset following 3 steps: firstly the individual sectoral water withdrawal is downscaled from country (or state) level to grid (0.5°x0.5°) level by using spatial downscaling algorithms; then annual time series of sector water withdrawal is obtained by using linear interpolation between the 5-year interval from reports; and finally a temporal downscaling procedure is adopted to generate monthly gridded water withdrawal data by sector. The sector-specific methodologies for the reconstruction of water withdrawal are described below in details.

### 2.2.1 Irrigation

Global gridded monthly irrigation water withdrawals during the period 1971-2010 are generated based on FAO AQUASTAT and USGS estimates and values of gridded monthly irrigation water withdrawals as simulated by four GHMs. Irrigation water withdrawals simulated by these four GHMs all have reasonable agreement (correlation coefficient ($r$) more than 0.7) with FAO AQUASTAT and USGS estimates at the country level and US state level, respectively (Figure S1). Large discrepancies exist among GHMs at the seasonal and regional scale (Figure S2) due to differences in model structure and parameters (Wada et al., 2013; Liu et al., 2017), so multiple GHMs are taken into account. By applying the correction factors between model estimates and reported estimates to the monthly gridded irrigation water withdrawals simulated by GHMs within a specific country (or state) (i.e. FAO AQUASTAT and USGS datasets), the reconstructed monthly gridded irrigation water withdrawal are calculated as follows:

$$Wir_{i,j,g} = Wir\_sim_{i,j,g} \times f_{m,p}, \qquad (1)$$

where $Wir_{i,j,g}$ is the reconstructed irrigation water withdrawal for the month $i$ of year $j$ at grid $g$ (m³), and $Wir\_sim_{i,j,g}$ is the irrigation water withdrawal for the month $i$ of year $j$ at grid $g$ simulated by four GHMs (m³); $f_{m,p}$ is the correction factor for the simulation by GHMs, calculated by $f_{m,p} = Wir\_obv_{m,p} / Wir\_sim_{m,p}$, where $Wir\_obv_{m,p}$ and $Wir\_sim_{m,p}$ are the 5-year irrigation water withdrawal (m³) reported by AQUASTAT (or USGS) and simulated by GHMs, respectively, for country (or state) $m$ (where grid $g$ is located in country $m$) and time period $p$ (year $j$ is in the period $p$). Thus, four reconstructed irrigation water withdrawal datasets are generated based on simulations from the four GHMs. The spatial and



temporal pattern of the ensemble mean of these four datasets, and the disagreement among them are discussed in results and discussion sections, respectively.

### 2.2.2 Domestic

The spatial downscaling of domestic water withdrawal follows the methods in Hejazi et al. (2014), which used the population
density maps as the proxy for disaggregating domestic water withdrawal from country (or state) level to grid level. Temporal downscaling algorithm for domestic water withdrawal are also used by Wada et al. (2011) and Voisin et al. (2013):

$$W_{dij} = \frac{W_{dj}}{12}\left(\frac{T_{ij}-T_{avg}}{T_{max}-T_{min}}R + 1\right), \qquad (2)$$

where $W_{dij}$ is domestic water withdrawal in month $i$ of year $j$ (m³); $W_{dj}$ is domestic water withdrawal in year $j$ (m³); $T_{ij}$ is the average temperature in month $i$ of year $j$; $T_{avg}$, $T_{max}$ and $T_{min}$ are the average, the maximum and the minimum monthly
temperature in year $j$ (all in °C), respectively; parameter $R$ is the amplitude (dimensionless), which measures the relative difference of domestic water withdrawal between the warmest and coldest months in a given year.

Wada et al. (2011) reported that $R$=0.1 could fit the variation of domestic water use in Japan and Spain. However, this term is different across regions as domestic water withdrawal is influenced not only by socioeconomic and climatic conditions but also by water policies and strategies (Babel et al., 2007). Here, we use the observed monthly water use data in 30 urban centers
and counties (Table 2) to calibrate $R$ in different regions. Table 3 shows the range of calibrated $R$ values for each country, and we use the median value for the temporal downscaling of domestic water withdrawal for the remaining countries with unavailable historical observation. Monthly domestic water withdrawal was calculated using Eq. (2) for the 30 urban centers and counties, and the simulated mean monthly domestic water withdrawal shows reasonable agreement with observations with correlation coefficient ($r$) more than 0.8 and mean absolute percentage error (MAPE) less than 15% in most urban centers and
counties (Fig. 1).

### 2.2.3 Electricity

Similar to the domestic sector, spatial downscaling of water withdrawal for electricity generation (water withdrawal for cooling of thermal power plants) is based on population density maps (Hejazi et al., 2014). The temporal downscaling of water withdrawal for electricity generation follows Voisin et al. (2013) and Hejazi et al. (2015), which assume that the amount of
water withdrawal for electricity generation is proportional to the amount of electricity generated. Here, the generated electricity is assumed to be consumed by three sectors, i.e., building, industry and transportation. Electricity consumption by building is further divided into three categories: heating, cooling and other home utilities. Electricity consumption for industry and transportation is assumed to be a uniformly distributed within a year, while water withdrawal for building electricity use is dependent on heating degree days (HDD) and cooling degree days (CDD). HDD and CDD, which are derived from outdoor
air temperature, are robust indicators for representing heating- and cooling-related energy consumption (Allen, 1976;





Karimpour et al., 2014). Here, only electricity use for heating and cooling are assumed to be sensitive to the climatic factors. Equation (3) represents for the temporal downscaling of electricity generation from annual to monthly:

$$E_{ij} = E_j \times (p_b \times (p_h \frac{HDD_{ij}}{\sum HDD_{ij}} + p_c \frac{CDD_{ij}}{\sum CDD_{ij}} + p_u \times \frac{1}{12}) + p_{it} \times \frac{1}{12}) \ , \tag{3}$$

where $E_{ij}$ is the electricity use for the month of $i$ and year of $j$; $E_j$ is the annual electricity use; $p_b$ and $p_{it}$ are the proportions of total electricity use for building and transportation and industry together, respectively, with $p_b + p_{it} = 1$; $p_h$, $p_c$ and $p_u$ are the proportions of total building electricity use for heating, cooling and other home utilities, respectively, with $p_h + p_c + p_u = 1$; $HDD_{ij}$ and $CDD_{ij}$ are the $HDD$ and $CDD$ of month $i$ in year $j$, respectively, and were calculated by using a base temperature of $18\,^{\circ}C$:

$$HDD_{ij} = \sum_1^n (18 - T_{d_{ij}}) \forall T_{d_{ij}} < 18\,°C \ , \tag{4}$$

$$CDD_{ij} = \sum_1^n (T_{d_{ij}} - 18) \forall T_{d_{ij}} > 18\,°C \ , \tag{5}$$

where $T_{d_{ij}}$ is the average temperature of the day $d$ of month $i$ in year $j$. Thus, the monthly water withdrawal for electricity generation is then calculated as follows:

$$W_{ij} = W_j \times (p_b \times (p_h \frac{HDD_{ij}}{\sum HDD_{ij}} + p_c \frac{CDD_{ij}}{\sum CDD_{ij}} + p_u \times \frac{1}{12}) + p_{it} \times \frac{1}{12}) , \tag{6}$$

where $W_{ij}$ is the water withdrawal of electricity generation for the month of $i$ and year of $j$; and $W_j$ is the annual total water withdrawal for electricity generation. The parameters $p_b$, $p_{it}$, $p_h$, $p_u$ and $p_c$ are obtained from the International Energy Agency (IEA) (IEA, 2012b, a). For some counties with low annual CDD (or HDD), there are almost no cooling (or heating) services. However, the parameters $p_c$ and $p_h$ (the proportions of total building electricity use for cooling and heating, respectively) are not equal to 0, which can lead to a failure in reproducing summer or winter peaks. Thresholds for annual HDD and CDD are defined, by assuming that if $\sum HDD_{ij} < 650\,°C$ or $\sum CDD_{ij} < 450\,°C$, then there is no electricity use for heating or cooling, respectively. Note, thresholds for annual HDD and CDD are obtained by calibration against reported monthly electricity generation data. The monthly water withdrawal for electricity generation is calculated as follows:

If $\sum HDD_{ij} < 650$ and $\sum CDD_{ij} < 450$:

$$W_{ij} = W_j \times \frac{1}{12} \ ; \tag{7}$$



If $\sum HDD_{ij} > 650$ and $\sum CDD_{ij} < 450$:

$$W_{ij} = W_j \times (p_b \times ((p_h + p_c)\frac{HDD_{ij}}{\sum HDD_{ij}} + p_u \times \frac{1}{12}) + p_{it} \times \frac{1}{12}) ; \qquad (8)$$

If $\sum HDD_{ij} < 650$ and $\sum CDD_{ij} > 450$:

$$W_{ij} = W_j \times (p_b \times ((p_h + p_c)\frac{CDD_{ij}}{\sum CDD_{ij}} + p_u \times \frac{1}{12}) + p_{it} \times \frac{1}{12}) ; \qquad (9)$$

If $\sum HDD_{ij} > 650$ and $\sum CDD_{ij} > 450$:

$$W_{ij} = W_j \times (p_b \times (p_h \frac{HDD_{ij}}{\sum HDD_{ij}} + p_c \frac{CDD_{ij}}{\sum CDD_{ij}} + p_u \times \frac{1}{12}) + p_{it} \times \frac{1}{12}) . \qquad (10)$$

Voisin et al. (2013) and Hejazi et al. (2015) validated this method against observed data for the year 2005 in US. To further validate this method globally, monthly electricity generation data during 2000-2012 in 33 OECD countries reported by IEA (http://www.iea.org/statistics/topics/Electricity/) were collected. Figure 2 shows the comparison between simulated and observed monthly mean electricity generation during 2000-2012 in 33 OECD countries. It is found that the simulations agree well (with the correlation coefficient above 0.6 and MAPE under 15%) with observations in most of the countries. However, electricity generation shows considerable underestimation in summer for some regions ( e.g. Austria, Chile, and Switzerland) where hydropower accounts for a large portion of the total electricity generations in summer and parts of electricity are exported to other countries (Bauer, 2009; Wagner et al., 2015; IEA, 2016). In general, the reasonable agreement between simulation and observation suggests the effectiveness of Eq. (7-10) to temporally downscale water withdrawal for electricity generation.

### 2.2.4 Livestock, mining and manufacturing

For the spatial downscaling, we apply the global maps of estimated livestock density to downscale water withdrawal of livestock (Alcamo et al., 2003; Hejazi et al., 2014), and population density to downscale water withdrawal of mining and manufacturing sectors. For the temporal downscaling of water withdrawal of livestock, mining and manufacturing, a uniform distribution (i.e. the monthly value are the same within the year) is adopted following Voisin et al. (2013).

### 3 Results

### 3.1 Spatial distribution of global water withdrawal by sectors



Figure 3 shows the spatial distribution of long-term mean annual water withdrawal by sector during 1971-2010. Total global water withdrawal has increased during the past 40 years,  and on average 68% of global water withdrawal has been used for irrigation, followed by electricity generation (11%), domestic (9%) and manufacturing (7%) while less than 5% of global total water withdrawal is for livestock and mining purposes. Irrigation water withdrawal is highest in the western US, eastern China, and India due to low water availability during the crop growing season and the massive crop productions in these regions. For example, in the western US, the average annual precipitation is less than 400 mm, resulting in water stress for optimal crop growth without irrigation. Different irrigation techniques for crops contribute to the large spatial heterogeneity of water withdrawal (Jägermeyr et al., 2015). For example, large amounts of water are withdrawn for maintaining a certain water level on rice fields in South China and Southeast Asia (Shahid, 2011). In addition, there is almost no irrigation in cold or sparsely populated regions (e.g. North Canada and Sahara). Domestic water withdrawals are high in the eastern US, eastern China, European countries, coastal regions of South America and India, but are limited in northern Canada, northern Russia and Sahara due to spare population. The spatial distributions of water withdrawal for electricity generation, mining and manufacturing are broadly similar to that of domestic, and consistent with the global population distribution that water withdrawal regions concentrating in urban areas or regions with denser population. As for the livestock sector, water withdrawal is mainly used in India, eastern China and the eastern US where livestock is densely concentrated (Robinson et al., 2014). Generally, the dominant water withdrawal sectors by land area are irrigation in the western US, eastern China, southern Brazil and India, domestic in the northern Brazil and most of the Africa, electricity generation in Russia, Canada, and the eastern US, and livestock in Australia (Fig.S3).

## 3.2 Seasonal patterns of water withdrawal for irrigation, domestic and electricity generation

An evident seasonal pattern is identified for irrigation water withdrawal during 1971-2010 (Fig. 4), concentrated in June to August (JJA) in the northern hemisphere and December to February (DJF) in the southern hemisphere. In the US and European countries, due to large water requirement in crop growing stages, more than 75% of annual irrigation water withdrawal occurs in JJA, while no irrigation takes place in DJF. In contrast, in the southern parts of South America and southern Africa, irrigation water is mainly withdrawn in DJF and accounts for about 70 percent of annual total irrigation. In general, irrigation water withdrawal exhibits an evident seasonal pattern in mid and high-latitude regions, but not in the tropical zone (e.g. Brazil and the Southeast Asia) where irrigation is applied year-around due mainly to multi-cropping practices. The seasonal variation of irrigation water withdrawal is determined not only by crop calendar but also the climate conditions. For example, in India, most of precipitation occurs in rainy seasons (monsoon) but crop water requirement is still large in September to November (SON), leading to a peak of irrigation water withdrawal in SON, especially in northwest India (Rodell et al., 2009; Famiglietti, 2014). The seasonal pattern of domestic water withdrawal (Fig. 5) is largely related to the seasonal temperature variation and the parameter $R$ (i.e. representing the relative difference of domestic water withdrawal between the warmest and coldest months). On both hemispheres, domestic water withdrawal is larger in the respective summer seasons compared to winter,




consistent with the seasonal evolution of temperatures. Water withdrawal for lawn and garden, which will take a large part of total domestic water withdrawal in summer, is the dominant factor for the summer peak, especially in developed countries (e.g. the US and Australia) (Loh and Coghlan, 2003; Shaffer, 2009). Figure 6 shows the seasonal pattern of water withdrawal for electricity generation. Higher water withdrawal is found in winter than in summer in high-latitude regions (e.g. Canada,

Western Europe and southern Australia), where heating is normally adopted in winter while cooling is rarely applied in summer time. On the contrary, electricity for heating is rarely used in winter in tropical regions (e.g. northern Africa and western Asia) as cooling is frequently applied in summer, resulting in dominant water withdrawal for electricity generation in summer. In fact, homes that have air conditions use electricity as the main source of cooling in the summer, while electricity is also one of the main sources for heating in winter (e.g. the application of furnace, boiler circulation pumps, and compressor) (EIA, 2017),

which leads to the summer and winter peak of electricity generation.

### 3.3 Trend in water withdrawal in 1971-2010 by sectors

Global total water withdrawal has increased significantly from 2500 to 4000 km$^3$ yr$^{-1}$ during 1971-2010 (Fig. S4). A particularly strong increasing trend is found in China (from ~400 to ~550 km$^3$ yr$^{-1}$) and India (from ~300 to ~800 km$^3$ yr$^{-1}$). In contrast, total water withdrawal in the US increased before 1980 but then decreased during 1985-2010, and similar evolution is found

for the European Union (EU27). Water withdrawal increased during the past 40 year in most regions (Fig. 7, Fig. S4-8) as a result of the increasing population, urbanization, the growing food demand and expansion of irrigated cropland, which are in line with previous studies (Shiklomanov, 2000; Wada and Bierkens, 2014). However, sectoral water withdrawal also shows decreasing trend in specific regions. Irrigation water withdrawal has exhibited a decreasing trend (about -0.3 mm/year) in western US and west Europe, partly due to the application of sprinkler and micro-irrigation systems (Pereira et al., 2002). A

significant decreasing trend of domestic water withdrawal is found in most of European countries (e.g. Sweden, Germany, and Poland), because of the low growth rate of population and the improvement of domestic water use efficiency and water management (e.g. water price and water meters) (Herrington, 1997; Gleick, 2000; Dalhuisen et al., 2003). In addition, in part of European countries and US, water withdrawal for electricity generation showed a decreasing trend, which could be attributed to shifts in cooling technologies and fuel mix. For instance, the penetration of more recirculating cooling technologies than

once-through, and the shift to less-water intensive fuel mixes (e.g., wind, solar, and natural gas) improved the overall water use efficiency of the electricity sector (Liu et al., 2015).

### 4 Discussion

The reconstructed global gridded monthly water withdrawal dataset by sector is generated by spatially and temporally downscaling country-scale estimates of sectoral water withdrawals from FAOSTAT (and state-scale estimates of USGS for





the US). In this section, the uncertainties in the data sources (FAO AQUASTAT and USGS) including model estimates, and in the applied spatial and temporal downscaling methods by sectors are discussed.

## 4.1 Uncertainties in data sources

Water withdrawal estimates by sectors in the US are provided by the USGS at a high spatial resolution (state and county), and are often treated as a benchmark for model calibration and validation (Vassolo and Döll, 2005; Hejazi et al., 2014; Leng et al., 2016). Water withdrawal estimates from FAO AQUASTAT are mainly from national surveys and assessments (e.g. national yearbook, statistics and reports) or model simulations (e.g. irrigation water withdrawal). Missing values in FAO AQUASTAT water withdrawal dataset were filled by Liu et al. (2016) with empirical techniques (e.g. population and irrigated area). Water withdrawals for electricity generation, mining and manufacturing were broken down from industrial estimates from FAO AQUASTAT with the aid of model simulations. Thus, uncertainties may arise from these procedures. To assess the level of uncertainty in the country-level data, we compared the domestic and industrial water withdrawal time series from 1971-2010 by  with estimates of Flörke et al. (2013) and Shiklomanov (2000) (Fig. S9). Global domestic water withdrawal agrees well among these estimates both in trend and average value. Global industrial water withdrawal estimates by Flörke et al. (2013) and Shiklomanov (2000) are higher than estimates used in this study, but they all show similar changing trend during 1970-2010. Estimates of thermoelectric water withdrawal in this study is lower than estimates from Flörke et al. (2013), and water withdrawal for manufacturing agrees well among these two datasets.

## 4.2 Uncertainties in reconstructed irrigation water withdrawal

The global gridded monthly irrigation water withdrawal data as produced in this study is based on various data sources, including both census national/state data and model estimates. Specifically, correction factors are used to adjust the irrigation water withdrawal estimates by GHMs to match the reported data at the country/state level. Therefore, besides the reliability of the data source, uncertainties among GHMs and different climate forcing would propagate into the newly developed dataset at the monthly time scale (Wada et al., 2013; Liu et al., 2017). Here, firstly four reconstructed irrigation water withdrawal datasets based on simulations of 4 GHMs, i.e. WaterGAP, H08, LPJmL, PCR-GLOBWB, forced by WFDEI, are compared to examine the uncertainties induced by model structure; then another four reconstructed irrigation water withdrawal based on simulations of WaterGAP forced by four climatic data, namely WFDEI, WATCH, GSWP3, Princeton, are used to investigate the uncertainties in reconstructed products induced by climate forcing. The coefficient of variation (CV) defined as the standard deviation divided by the ensemble mean value of these four generated datasets are used to evaluate the uncertainty. As shown in Fig. 8, the uncertainties arising from GHMs are rather high (CV>0.5) in the southeast China, the west coast of South America, the southeast of Brazil and part of the US. Seasonally, CVs in the northern hemisphere are larger than these in the southern hemisphere in DJF and vice versa in JJA (Fig. S10). Uncertainties among GHMs in irrigation water withdrawal simulation mainly come from the parameterization and assumptions of irrigation scheme, such as the crop calendar, irrigation



area and crops types (Wada et al., 2016). Although all four GHMs rely on approximately the same data set of irrigated areas from Siebert et al. (2005) (GMIA, http://www.fao.org/nr/water/aquastat/irrigationmap/index.stm), the crop types and the crop calendar definition in these GHMs are different. For example, LPJmL, H08 and WaterGAP use climate conditions to simulate crop calendars (Bondeau et al., 2007; Hanasaki et al., 2010), while PCR-GLOBWB use the crop calendar data from Portmann

et al. (2010). In addition, the uncertainty arising from climate forcing is small in most of regions (CV<0.25) due to the high agreement of historical climate datasets (Müller Schmied et al., 2016). Therefore, it is evident that the uncertainty from model structure is larger than that induced by forcing data. To improve the reconstruction of irrigation water withdrawal data, more realistic irrigation parameterization in GHMs and more reliable input data are needed.

## 4.3 Uncertainties in the spatial and temporal downscaling methods

Although the applied spatial and temporal downscaling methods possess some level of uncertainty in how water withdrawals are distributed spatially within a region or within a year, we did not explore the role of different downscaling methods on the gridded water withdrawal results. Instead we relied on a set of methods that have been used in the literature (Wada et al., 2011; Voisin et al., 2013; Hejazi et al., 2014; Wada and Bierkens, 2014) due to the general lack of multiple methods. Thus, we limit our discussion here to some of the potential sources of uncertainties associated with the spatial and temporal downscaling

methods.

The spatial downscaling of water withdrawal by sectors can benefit from considering additional factors to represent the spatial distribution of global water withdrawal. For example, the spatial distribution of domestic water withdrawal is related not only to population density but also to incomes (GDP per capita) (Flörke et al., 2013), which varies region by region. In addition, water withdrawal for electricity generation is mainly for cooling purpose in thermoelectric power plant, and can also be affected

by many factors, including the location of power plants, the amount of generated electricity, generation type, cooling technology, and fuel type (Byers et al., 2014; Hejazi et al., 2014; Liu et al., 2015). As for mining and manufacturing sectors, Vassolo and Döll (2005) found that the consideration of city nighttime lights works better that urban population. In addition, water withdrawals for manufacturing and mining are also dependent on the purpose for water use (e.g. cleaning and cooling), the outputs type (e.g. food and beverages), GDP, and the technical system of water use (Flörke et al., 2013). Thus, future

research should also consider using other ancillary data in addition to population density maps for the spatial downscaling of domestic and industry water withdrawals, such as the geographic locations and characteristics of power plants, manufacturing centers, and mines, and their historical evolutions.

The temporal downscaling methods by sectors can benefit from accounting for the intra-seasonal and inter-annual pattern of water withdrawal. That is the inter-annual variation of water withdrawal by sectors need to be considered when downscaling

FAO AQUASTAT and USGS data from of 5-yr interval to annual scale. The inter-annual variability of human water withdrawal is of great significance for understanding the impacts of climate change (e.g. El Niño-Southern Oscillation, drought, and flood) on human behavior and economy (Vörösmarty et al., 2000; Jacob, 2001; Piao et al., 2010; Haddeland et



al., 2014). Furthermore, temporal downscaling of domestic water withdrawal can benefit from considering additional factors besides air temperature , such as precipitation, population, and water availability to represent the seasonality of domestic water withdrawal (White et al., 1972; Hoekstra and Chapagain, 2006). Also, observed monthly domestic water withdrawal data will be of great importance for the calibration and validation of the parameter $R$ in regions without historical observations. As for

electricity generation, the effects of electricity trade and hydropower generation need to be taken into account in future research. Although air temperature datasets used for temporal downscaling may add another source of uncertainty to the reconstructed water withdrawal data, our results show that the uncertainty induced by air temperature datasets is small in the temporal downscaling of water withdrawal for domestic and electricity generation (Fig.S11). This is mainly because of the high agreement in monthly variation of air temperature among the four different data sources (i.e. WFDEI, WATCH, GSWP3,

Princeton) as all of them are bias corrected to (different) versions of the CRU time series (Müller Schmied et al., 2016).

## 5 Conclusions

In this study, a reconstructed global gridded monthly sectoral water withdrawal dataset was produced for the period 1971-2010 by temporally and spatially downscaling country-level (FAO AQUASTAT) and state-level (USGS, only for USA) datasets using various models and new modeling approaches. Correction factors are used to scale irrigation water withdrawal estimates

by GHMs to annual country/state estimates from FAO and USGS. Global population density maps are used for the spatial downscaling for water withdrawal for domestic, electricity generation, mining and manufacturing; while livestock density maps are used for livestock sector. In addition,  air temperature are used to present the monthly variation of water withdrawal by domestic and electricity generation, which are validated against observations, and simulation results show reasonable agreements with observations in selected regions.

The reconstructed dataset, at 0.5 degree spatial resolution and monthly temporal resolution, includes water withdrawal by sector, i.e. irrigation, domestic, electricity generation, livestock, mining and manufacturing. Based on the reconstruction dataset, the spatial and temporal change patterns of global water withdrawal by sectors were analyzed. Globally, most of global water withdrawal is used for irrigation, followed by electricity generation and domestic. Spatially, the dominant irrigation water withdrawal area are regions with large irrigated cropland and massive crop productions, e.g. the western US, eastern

China, and India. Water withdrawal for domestic, electricity generation, mining and manufacturing are high in urban areas or regions with denser population. Seasonally, irrigation water withdrawal exhibits an evident seasonal pattern in mid and high-latitude regions, but not in the tropical zone.  Domestic water withdrawal is larger in JJA than in DJF in northern hemisphere, and vice versa in southern hemisphere. Water withdrawal for electricity generation showed a winter peak in high-latitude regions and a summer peak in low-latitude regions.

In addition, the uncertainties in the reconstructed water withdrawal data are analyzed, and limitation for spatial and temporal downscaling of other sector are discussed. Results show that the uncertainties arising from model structure are larger than that induced by forcing data in the reconstructed irrigation water withdrawal. More advanced models that capture the spatial pattern



and intra- and inter-annual variabilities of sectoral water withdrawal are prospect, and more frequently and spatially resolved observed water withdrawal data at country or region scale are also required for improving the quality of the reconstructed dataset. In whole, despite the uncertainties and limitations, this study is of great significance not only for cross-comparison and validation for modeling and analyzing the impacts of human water use, but also for investigating water use related issues

at finer spatial, temporal and sectoral scales.

## Acknowledgement

This research was supported by the Office of Science of the US Department of Energy through the Integrated Assessment Research Program. PNNL is operated for DOE by Battelle Memorial Institute under contract DE-AC05-76RL01830. Zhongwei Huang would like to acknowledge support from the National Natural Science Foundation of China (41425002) and

the National Youth Top-notch Talent Support Program in China.

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



**Table 1 Datasets for spatial and temporal downscaling of reported water withdrawal by sectors**

| Sectors | Spatial downscaling | Temporal downscaling |
|---|---|---|
| Irrigation | Global irrigation water withdrawal simulation by 4 GHMs (namely WaterGAP, H08, LPJmL, and PCR-GLOBWB) for the period 1971-2010 | |
| Domestic Electricity | Global population density maps from HYDE during 1970-1980 and GPW during 1985-2010 | The gridded daily air temperature data from WFDEI during 1971-2010 |
| Mining Manufacturing | | uniform distribution |
| Livestock | Global livestock density maps in 2005 from FAO | uniform distribution |



**Table 2 Details of the observed monthly domestic water withdrawal for calibration of parameter R.**

| Country | City | Period | Source |
|---------|------|--------|--------|
| Canada | Kindersley | 2001-2015 | Saskatchewan  community water use records, Water security agency(2016) |
| | Assiniboia | 2001-2015 | |
| | Yorkton | 2001-2015 | |
| | Prince Albert | 2003-2015 | |
| | Stanley Mission | 2005-2014 | |
| | Estevan | 2001-2015 | |
| | Swift Current | 2001-2015 | |
| | Estend | 2001-2015 | |
| | Regina | 2001-2015 | |
| USA | Indiana | 1999-2004 | Shaffer (2009) |
| | Ohio | 1999-2004 | |
| | Canyon | 1971-1978 | Maidment and Parzen (1984) |
| | Phoenix | 1995-2004 | Balling et al. (2008) |
| | Tucson | 1990 | Voisin et al. (2013) |
| | Seattle | 1990 | |
| | Orange | 1990 | |
| | Clemson University | 1990 | |
| | Fortuna | | State Water Resources Control Board of California (http://projects.scpr.org/applications/onthly-water use/) |
| | Imperial | | |
| | Galt | | |
| | Ripon | | |
| | Greenfield | 2013, 2015 | |
| | Riverbank | | |
| | Truckee-Donner | | |
| | Fillmore | | |
| | Hanford | | |
| | Adelanto | | |
| India | West Bengal | 2006 | Hossain et al. (2013) |
| China | Beijing | 2013-2014 | Beijing Water Authority (https://www.bjwater.gov.cn/pub/bjwater/bmfw/) |
| Australia | Perth | 2000-2001 | Loh and Coghlan (2003) |





**Table 3 Calibrated R in different counties and their median value for temporal downscaling of domestic water withdrawal.**

|  | Canada | USA | Australia | India | China | Japan | Spain | Global |
|---|---|---|---|---|---|---|---|---|
| City number | 9 | 18 | 1 | 1 | 1 | 1 | 1 | 32 |
| Range of R | 0.15~0.79 | 0.11~1.14 | - | - | - | - | - | 0.1~1.14 |
| Median R | 0.36 | 0.52 | 0.8 | 0.29 | 0.2 | 0.1 | 0.1 | 0.45 |





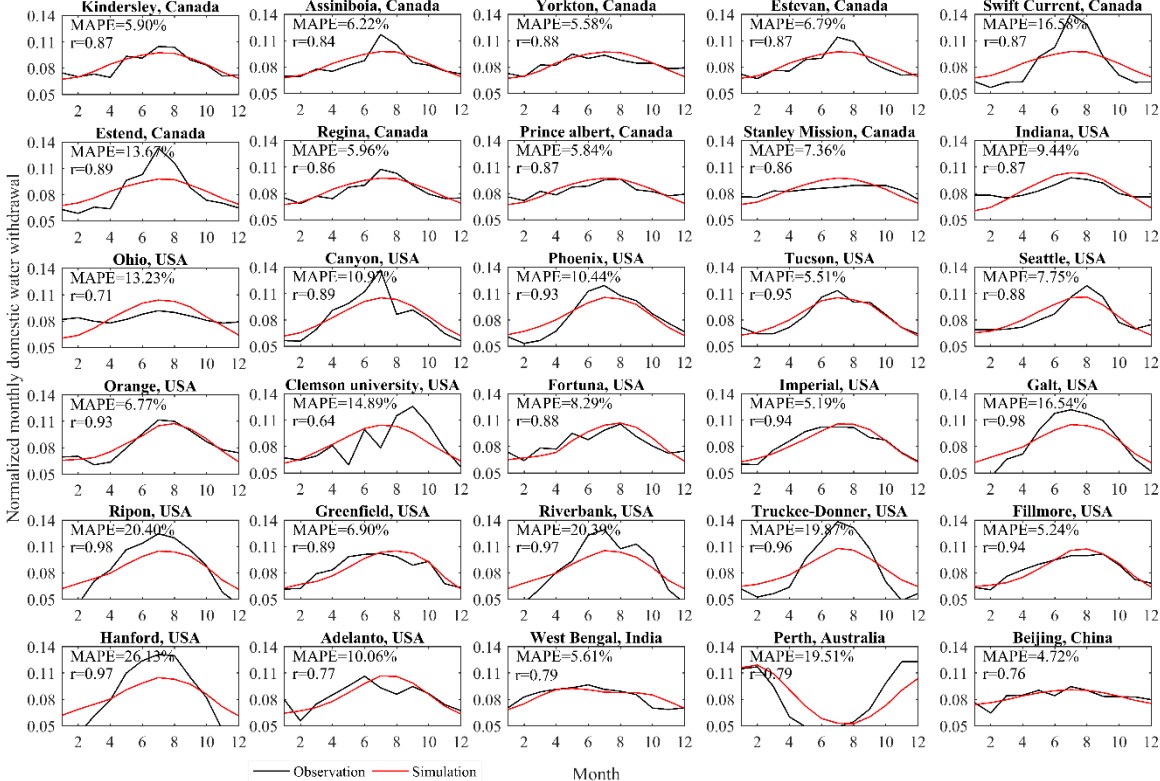

**Figure 1 Comparison between simulated and observed monthly domestic water withdrawal in global 30 regions: the normalized monthly water withdrawal is the proportion of monthly water withdrawal to the total annual water withdrawal.**





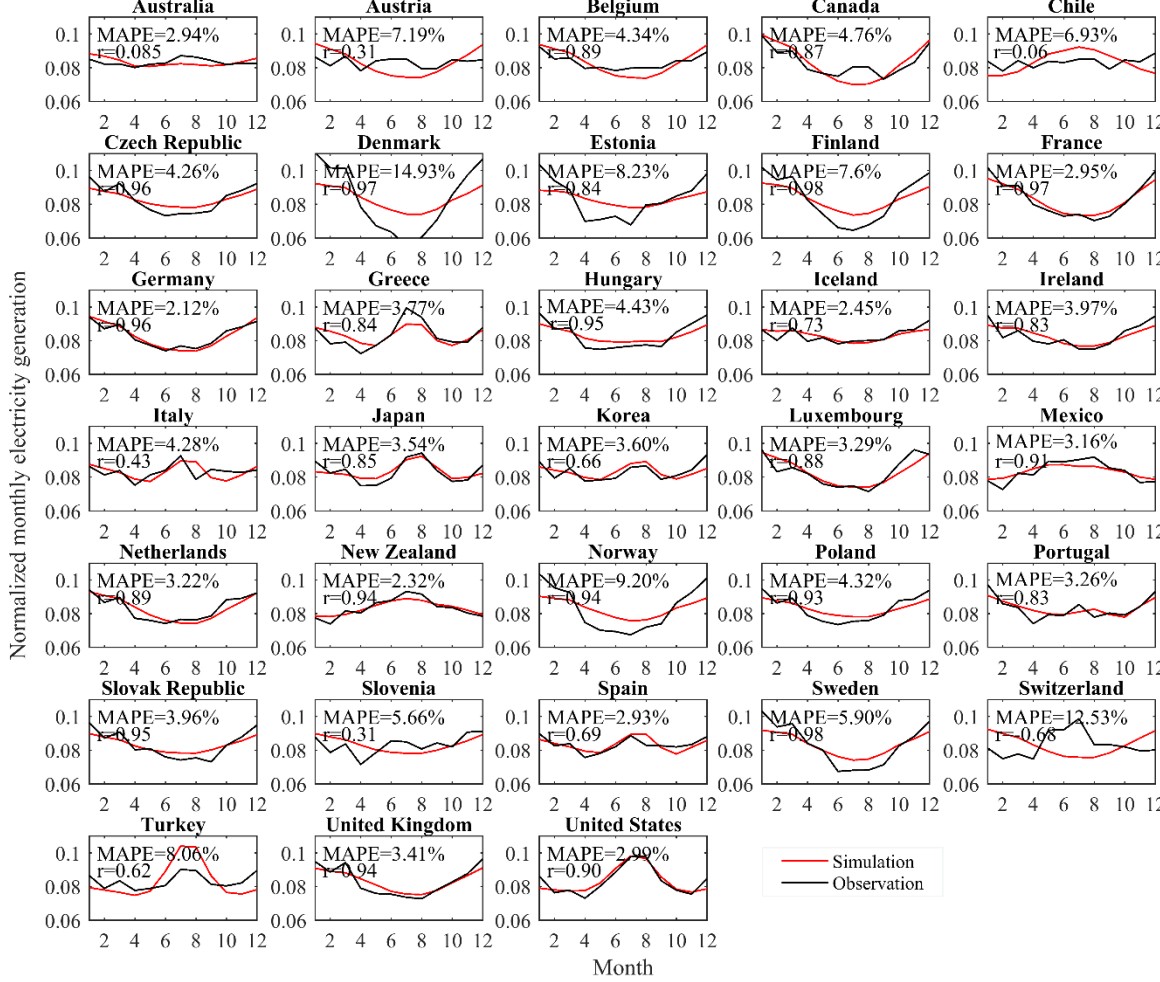

**Figure 2 Comparison between simulation and observation of normalized monthly mean electricity generation in 33 OECD countries during 2000-2012: the normalized monthly electricity generation is the proportion of monthly water withdrawal to the total annual electricity generation.**



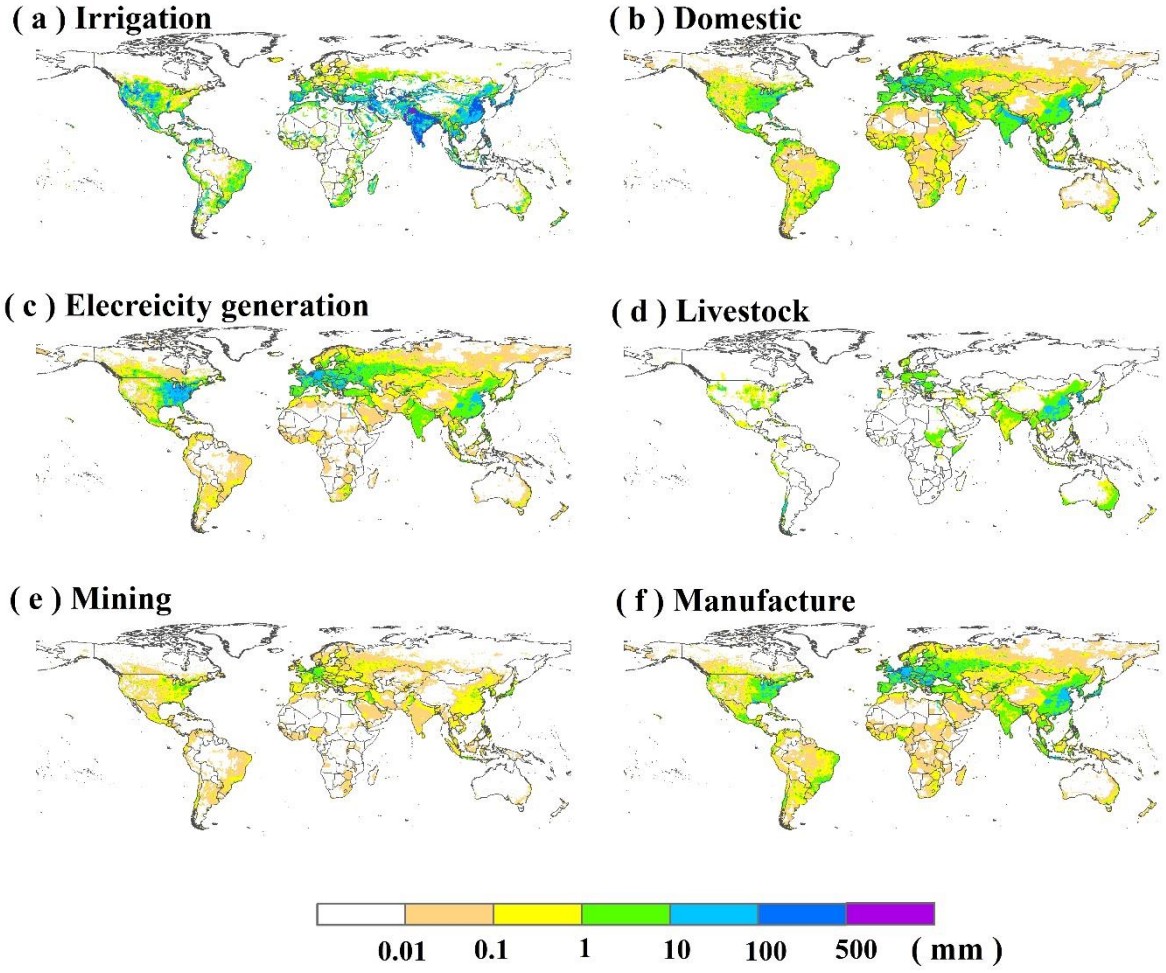

**Figure 3 Spatial distribution of annual mean water withdrawal by 6 sectors: (a) irrigation, (b) domestic, (c) electricity generation, (d) livestock, (e) mining and (f) manufacturing during 1971-2010.**




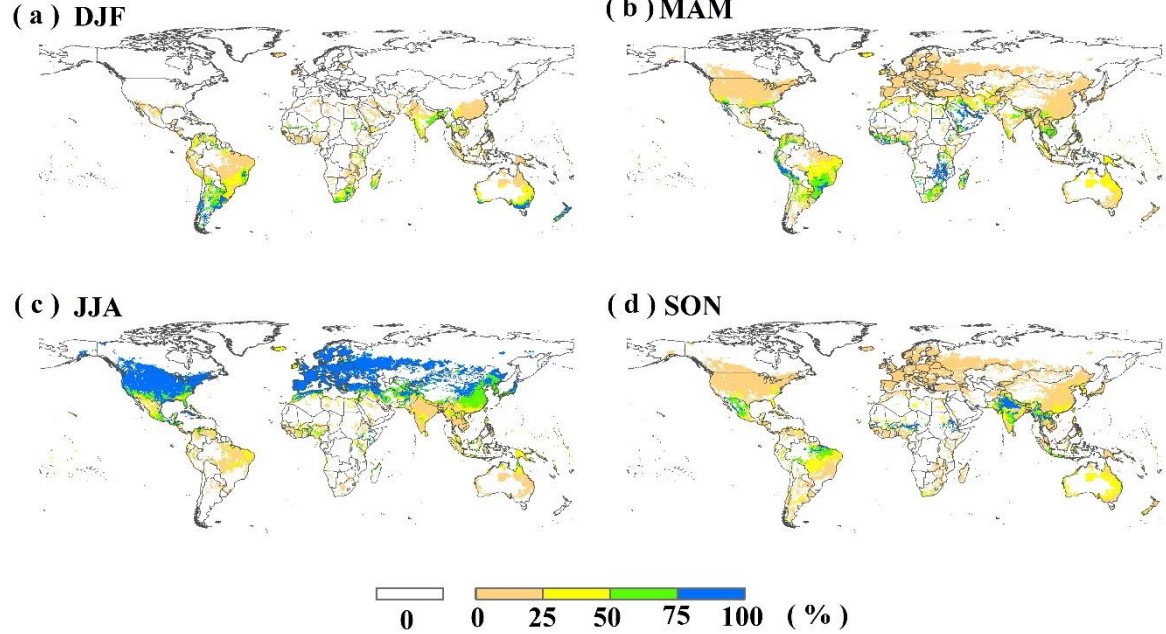

**Figure 4 Relative seasonal distribution of global irrigation water withdrawal over the period 1971-2010 based on the ensemble mean of four GHMs: December to February (DJF), March to May (MAM), June to August (JJA) and September to November (SON).**



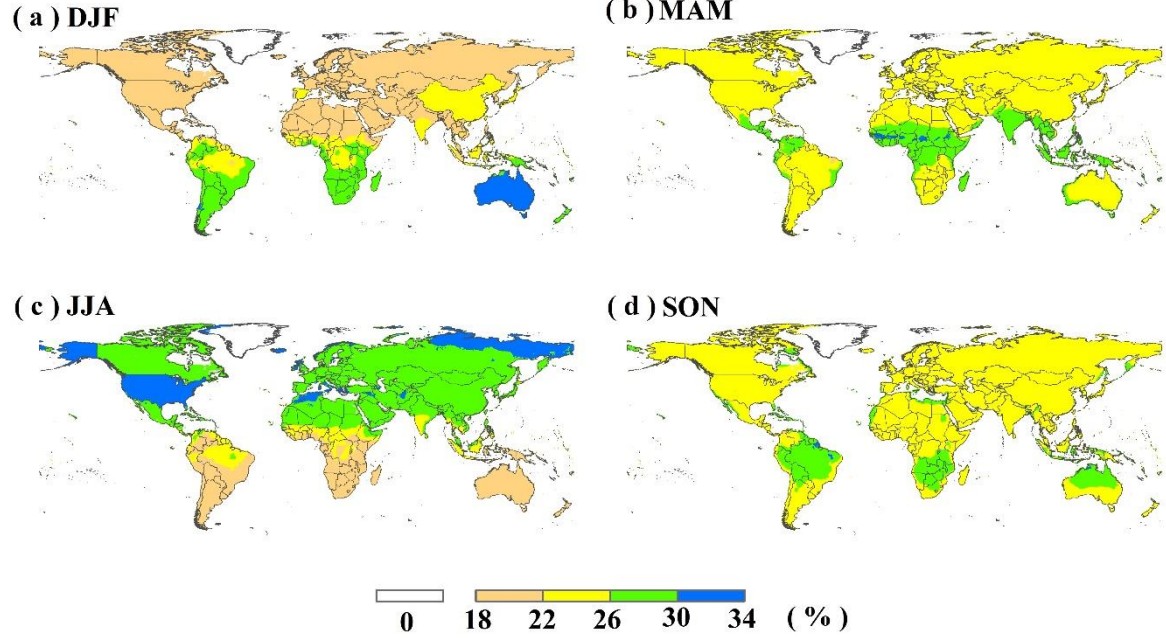

**Figure 5 Relative seasonal distribution of global domestic water withdrawal over the period 1971-2010: December to February (DJF), March to May (MAM), June to August (JJA) and September to November (SON).**





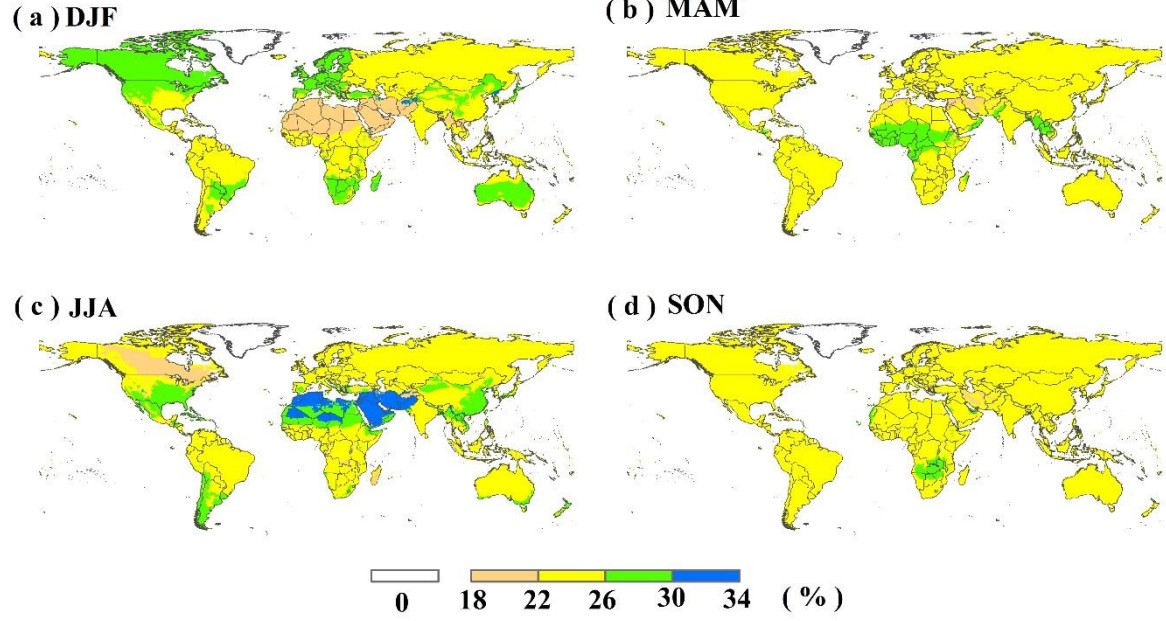

**Figure 6 Relative seasonal distribution of global electricity generation water withdrawal over the period 1971-2010: December to February (DJF), March to May (MAM), June to August (JJA) and September to November (SON).**





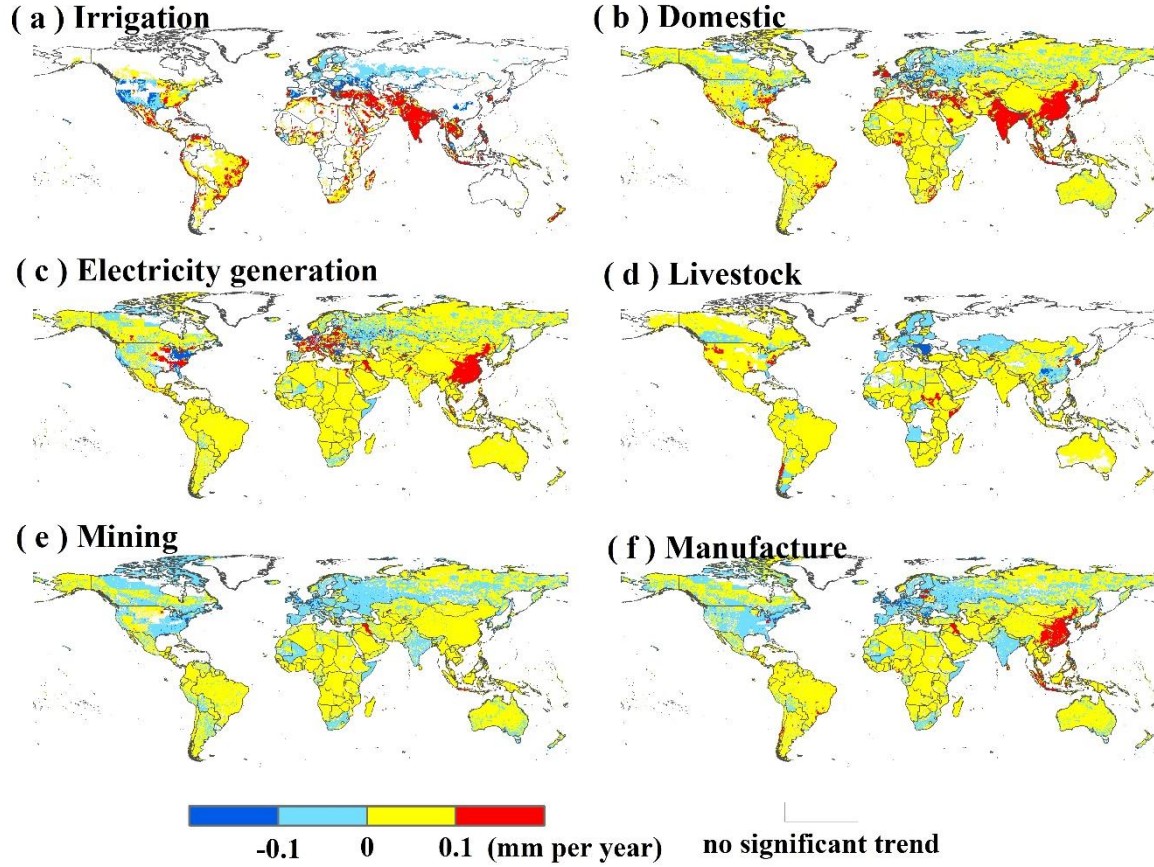

**Figure 7 Trend of global gridded water withdrawal by sectors: (a) irrigation, (b) domestic, (c) electricity generation, (d) livestock, (e) mining and (f) manufacturing.**





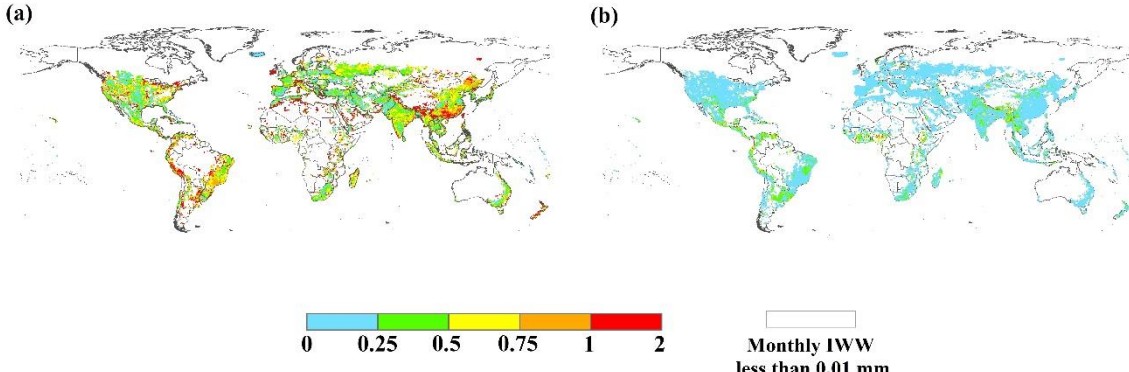

**Figure 8 Coefficient of variation (CV) in multi-annual average irrigation water withdrawal caused by (a) multi-model framework and by (b) multi-forcing data, and area with annual irrigation water withdrawal (IWW) less than 0.01 mm are not taken into consideration.**

