# Peer review of "Reconstruction of global gridded monthly sectoral water withdrawals for 1971-2010 and analysis of their spatiotemporal patterns"

_Hydrology and Earth System Sciences, 2017_

## Referee Comment (RC1) · Anonymous Referee #1 · 8 Jan 2018

In this study, the authors reconstructed the global water withdrawal patterns from collected data by statistical downscaling. The spatial and temporal patterns of water withdrawal, along with sectoral divisions were analyzed. This work is not trivial. Estimating water withdrawal in a small watershed and considering various sectors is hard enough, not to say at the global scale. As a result, I do not think readers should blame the simplifications taken here.

However, I do have a concern of the irrigation part. It seems that the observations used for calibration is very sparse, especially in developing countries. For example, in the two major countries with water withdrawal – China and India, only data from West

Bengal and Beijing were used. The result might be very biased because of the spatial variability of climate, water resources, and population density. Considering that 68% of water withdrawal is used for irrigation, this might lead to large errors in the final result.

Also in Table 2: The second column is a mixture of cities, counties and states. In addition, it is better to indicate which state the city is located as it is not uncommon for multiple cities to have the same name.

In Table 3, did you calibrate the R value in Japan and Spain too, or you just adopted the value from literature?

Overall, the study is novel, the topic is suitable to HESS, and the manuscript is well written. I suggest a minor revision addressing my concerns mentioned above.

---

## Referee Comment (RC2) · Anonymous Referee #2 · 10 Jan 2018

GENERAL COMMENTS

This manuscript reconstructs gridded monthly water withdrawals globally for 6 sectors for 1971-2010 in a spatial resolution of 0.5*0.5 degrees. The authors make this water withdrawal dataset publicly available, which makes the manuscript more valuable and is in the line of the open-access philosophy of HESS.

Such a detailed global dataset is indeed to my knowledge the first in its kind and very useful. The statement at the end of the document (page 14 lines 3-5) "In whole, despite the uncertainties and limitations, this study is of great significance not only for cross-comparison and validation for modeling and analyzing the impacts of human water use,

but also for investigating water use related issues at finer spatial, temporal and sectoral scales" is very true.

I also appreciate that the authors include an extensive part in their manuscript on uncertainty (Section 4), as they acknowledge the uncertainty and limitations of their study.

The manuscript is novel, well written and in the scope of HESS. I recommend for moderate revision, as some issues need to be additionally addressed/discussed first.

MODERATE COMMENTS

1) The authors use as basis FAO AQUASTAT data and state-scale estimates of USGS for the US as basis for downscaling. Yet, on page 3 Lines 1-15 they argue that particular countries provide more detailed (especially spatially) data than the FAOSTAT data. This is indeed true for Germany as the authors point out, but also for many other European countries. These data (and additionally from Canada, China, ...) could have been used to optimise the downscaling methodology the authors use. Why was this choice made for the US but not for these other sources? I find this a bit a missed opportunity. I acknowledge that this means a lot more work, but you could have used all best data available instead of the US selection. Nevertheless, this does not have to be done within this paper, but maybe in future work. Please discuss shortly in the limitations section (section 4) of your manuscript.

2) SPATIAL DOWNSCALING TECHNIQUES: For some sectors (domestic, irrigation, livestock) the downscaling techniques are state of the art, for other sectors (electricity for cooling, mining and manufacturing) they are very rough. The three latter are based upon population-density maps. This is a very rough approach, as these sectors are in my opinion not always highly correlated with population densities. Water abstractions for cooling can very well be concentrated outside urban centres, for security reasons (e.g. nuclear power plants) and the availability of large water quantities (e.g. along rivers). Nulear water abstraction whcih can be substantial can thus be concentrated as point intake in a more rural area. Manufacturing industries have in developed countries

often moved outside urban centres (where in the past they were often in city centres). Last but not least, mining activities often take place in remote areas, and large water abstractions can be very concentrated on a small rural spatial scale. When you produce a 0.5*0.5 degree geodataset, these considerations can be very relevant. I acknowledge that the authors briefly describe limitations on page 12 lines 24-27. They also say this is a topic for further research. But please elaborate more on this, in the line with the argumentation I just made.

3) MISSING SECTOR TOURISM: The authors include 6 sectors, the ones which are typically identified for abstracting water. However, as in most studies, some particular water abstraction sectors are excluded. As indicated in the publication https://doi.org/10.1016/j.ecoser.2015.08.003 , an important generally neglected sector is tourism. This includes water abstractions for snowmaking, which during winter months in mountain areas can be the largest regional water user (https://doi.org/10.2166/wst.2009.211 ). This water is generally taken from surface water, and is not accounted for in municipal water abstraction statistics. But this also includes water abstractions for hotels/swimming pools/spas both in winter and summer tourist areas (e.g. https://doi.org/10.1016/j.tourman.2013.05.010 ). These water users often have own private water abstractions, which are not accounted for in domestic/municipal water use statistics. E.g. in Mediterranean regions during summer months these water abstractions can become shortly the dominant water use. Another touristic water user are golf courts (e.g. https://doi.org/10.1094/ATS-2009-0129-01-RS). These touristic water abstractions can on a local (0.5*0.5 degree) and temporal (monthly) level be very significant. Please include in your discussion section a short subsection on this topic, based upon my input. Future research should include the sector tourism.

4) SECTORS FORESTRY and AQUACULTURE: As indicated in the publication https://doi.org/10.1016/j.ecoser.2015.08.003 , these sectors also account for water abstractions. Again, on a global level they may not be very significant in quantity, but on
a local (0.5*0.5 degree) and temporal (monthly) level, they can be very significant. Is forestry accounted for in your irrigation sector? Aquaculture can be very significant in a country like China. Please include in your discussion section a short subsection on this topic.

5) DOMESTIC WATER ABSTRACTION: Please define in your paper what you mean with this. There is often confusion in the terminologies domestic and municipal water abstractions. There is a difference in water abstractions by households (generally defined as domestic water abstractions) and municipal water use, which additionally includes water use by shops, schools, public buildings ... and even for the cleaning of streets or public parks. As I understand your definition of "domestic sector" also includes these water users. Include a definition.

MINOR COMMENTS

Page 2 line 19: You discussed the impact on the hydrological cycle and humans. Please add a sentence about the negative impact on the environment

page 2 Line 22 'We focus in this study on water withdrawal" - This is a choice, as also water consumption is an important statistic of water use. Water stress e.g. can be computed with both, as discussed in a recent publication https://doi.org/10.1016/j.scitotenv.2017.09.056

Page 3 line 7: Please add that also other selected European countries provide more detailed water use statistics (especially spatial data)

Page 3 Lines 18, 19: GHM and LSM - define abbreviation first

Page 4 Line 14 ... (GCAM): please add ref

Page 6 Line 14: ... 30 urban centers ... : Urban water use characteristics can actually be quite different from rural water characteristics. By only down scaling based upon urban water use characteristics, the resulting dataset could be biased in temporal representation for more rural areas

Page 8 Lines 17-21: Water abstraction for livestock: there are actually formulas that relate livestock water use to temperature.

Table 1: Please add a column with the spatial resolution of these datasets

Figure 3: (c) Electricity and not elecreicity.

---

## Referee Comment (RC3) · Anonymous Referee #3 · 24 Jan 2018

This manuscript aims to reconstruct a global monthly gridded (0.5 degree) sectoral water withdrawal dataset for six water use sectors (irrigation, domestic, electricity generation (cooling of thermal power plants), livestock, mining, and manufacturing) for the period 1971-2010. And the reconstructed gridded water withdrawal dataset is open-access. This paper is suitable for the HESS scope and also a valuable contribution to examining issues related to water withdrawals at fine spatial, temporal and sectoral scales.

The spatial distribution of water withdrawal for electricity generation depends on the distribution of the power plants. Most of the power plants are not concentrated in densely

populated area. However, in this paper, spatial downscaling of water withdrawal for electricity generation (water withdrawal for cooling of thermal power plants) is based on population density maps. It should be future explained and discussed.

In this paper, the spatial downscaling of water withdrawal for water withdrawal of electricity generation, domestic, mining and manufacturing was based on the population density maps. According to the gridded population map of the world (Center for International Earth Science Information Network (CIESIN) Columbia University), there are no people in Taklimakan Desert, some "no man's land" areas in Qinghai-Tibet Plateau, Sahara Desert. However, there are some water withdrawal of those sectors (please see Figure 5, 6, and 7). And in Figure S3, the dominant water withdrawal sector is manufacturing in Taklimakan Desert and some "no man's land" areas in Qinghai-Tibet Plateau, and is domestic in Sahara Desert. Please check it.

———————————————

---

## Author Comment (AC1) · 2 Feb 2018

**A point-by-point response to the reviews for "Reconstruction of global gridded monthly sectoral water withdrawals for 1971–2010 and analysis of their spatiotemporal patterns" by Zhongwei Huang et al**

Manuscript Details: Reconstruction of global gridded monthly sectoral water withdrawals for 1971-2010 and analysis of their spatiotemporal patterns, https://doi.org/10.5194/hess-2017-551

Authors: Zhongwei Huang, Mohamad Hejazi, Xinya Li, Qiuhong Tang, Chris Vernon, Guoyong Leng, Yaling Liu, Petra Döll, Stephanie Eisner, Dieter Gerten, Naota Hanasaki, Yoshihide Wada

We thank the reviewer for the very valuable comments and suggestions to improve the manuscript. Our point-by-point responses are listed below.

**Response to Anonymous Referee #1**

*Referee comments in Italics*

*In this study, the authors reconstructed the global water withdrawal patterns from collected data by statistical downscaling. The spatial and temporal patterns of water withdrawal, along with sectoral divisions were analyzed. This work is not trivial. Estimating water withdrawal in a small watershed and considering various sectors is hard enough, not to say at the global scale. As a result, I do not think readers should blame the simplifications taken here.*

Response: We appreciate the positive and constructive feedback from the referee on our manuscript.

*However, I do have a concern of the irrigation part. It seems that the observations used for calibration is very sparse, especially in developing countries. For example, in the two major countries with water withdrawal – China and India, only data from West Bengal and Beijing were used. The result might be very biased because of the spatial variability of climate, water resources, and population density. Considering that 68% of water withdrawal is used for irrigation, this might lead to large errors in the final result.*

Response: We agree with the reviewer that irrigation is the largest water withdrawer and consumer of water globally and certainly in countries such as China and India. The mentioned example of using sparse data was specifically for the temporal downscaling of domestic water withdrawals, but not for irrigation. For irrigation, we used the gridded irrigation water withdrawal estimates from Global Hydrological Models (GHMs) as a base layer to spatially and temporally downscale the reported country-level irrigation data from FAO AQUASTAT and USGS. As for the domestic sector, we collected monthly domestic water withdrawal from various sources (Table 2) to guide the temporal downscaling of domestic water withdrawals. As the referee mentioned, in the two major countries with water withdrawal – China and India, only data from West Bengal and Beijing were used. Given that domestic water withdrawal is roughly 7% of total water withdrawal in India and 12% in China, we acknowledge that more data would help improve the temporal downscaling of domestic water withdrawals, and future work should focus on collecting high resolution water withdrawal data both spatially and temporally. In the revised version of our manuscript, we have discussed these aspects in detail.

*Also in Table 2: The second column is a mixture of cities, counties and states. In addition, it is better to indicate which state the city is located as it is not uncommon for multiple cities to have the same name.*

Response: Thanks for your kind comment. We have revised Table 2 as suggested.

*In Table 3, did you calibrate the R value in Japan and Spain too, or you just adopted the value from literature?*

Response: We adopted the R value in Japan and Spain from the literature because we don't have monthly water withdrawal data for these two countries. We have clarified this distinction in the revised manuscript.

*Overall, the study is novel, the topic is suitable to HESS, and the manuscript is well written. I suggest a minor revision addressing my concerns mentioned above.*

Response: Thanks to the referee for the positive feedback. We have revised our manuscript based on the suggestions and comments.

---

## Author Comment (AC2) · 2 Feb 2018

**A point-by-point response to the reviews for "Reconstruction of global gridded monthly sectoral water withdrawals for 1971–2010 and analysis of their spatiotemporal patterns" by Zhongwei Huang et al**

Manuscript Details: Reconstruction of global gridded monthly sectoral water withdrawals for 1971-2010 and analysis of their spatiotemporal patterns, https://doi.org/10.5194/hess-2017-551

Authors: Zhongwei Huang, Mohamad Hejazi, Xinya Li, Qiuhong Tang, Chris Vernon, Guoyong Leng, Yaling Liu, Petra Döll, Stephanie Eisner, Dieter Gerten, Naota Hanasaki, Yoshihide Wada

We thank the reviewer for the very valuable comments and suggestions to improve the manuscript. Our point-by-point responses are listed below.

**Response to Anonymous Referee #2**

*Referee comments in Italics*

*GENERAL COMMENTS*

*This manuscript reconstructs gridded monthly water withdrawals globally for 6 sectors for 1971-2010 in a spatial resolution of 0.5\*0.5 degrees. The authors make this water withdrawal dataset publicly available, which makes the manuscript more valuable and is in the line of the open-access philosophy of HESS.*

*Such a detailed global dataset is indeed to my knowledge the first in its kind and very useful. The statement at the end of the document (page 14 lines 3-5) "In whole, despite the uncertainties and limitations, this study is of great significance not only for cross comparison and validation for modeling and analyzing the impacts of human water use, but also for investigating water use related issues at finer spatial, temporal and sectoral scales" is very true.*

*I also appreciate that the authors include an extensive part in their manuscript on uncertainty (Section 4), as they acknowledge the uncertainty and limitations of their study.*

*The manuscript is novel, well written and in the scope of HESS. I recommend for moderate revision, as some issues need to be additionally addressed/discussed first.*

Response: We appreciate the positive and constructive feedback from the referee on our manuscript.

*MODERATE COMMENTS*

*1) The authors use as basis FAO AQUASTAT data and state-scale estimates of USGS for the US as basis for downscaling. Yet, on page 3 Lines 1-15 they argue that particular countries provide more detailed (especially spatially) data than the FAOSTAT data. This is indeed true for Germany as the authors point out, but also for many other European countries. These data (and additionally from Canada, China, ...) could have been used to optimise the downscaling methodology the authors use. Why was this choice made for the US but not for these other sources? I find this a bit a missed opportunity. I acknowledge that this means a lot more work, but you could have used all best data available instead of the US selection. Nevertheless, this does not have to be done within this paper, but maybe in future work. Please discuss shortly in the limitations section (section 4) of your manuscript.*

Response: Thanks for your thoughtful comments. We agree with the reviewer that we could have improved the spatial downscaling if we collected subnational sectoral water withdrawals for these countries. We also agree with the reviewer that such an extension would amount to a lot of additional work and should be tackled in future research. Such an effort would also raise some additional challenges. For example, the definitions of sectoral water use are potentially inconsistent because these data are reported by various organizations and institutions. We only use FAOSTAT and USGS data in this study, but we can update the open-access datasets after we obtain the subnational sectoral water withdrawal data for other regions or countries. In the revised manuscript, we have further discussed the limitations of this work and potential future work needed to improve the reconstructed dataset.

*2) SPATIAL DOWNSCALING TECHNIQUES: For some sectors (domestic, irrigation, livestock) the downscaling techniques are state of the art, for other sectors (electricity for cooling, mining and manufacturing) they are very rough. The three latter are based upon population-density maps. This is a very rough approach, as these sectors are in my opinion not always highly correlated with population densities. Water abstractions for cooling can very well be concentrated outside urban centres, for security reasons (e.g. nuclear power plants) and the availability of large water quantities (e.g. along rivers).*

*Nulear water abstraction which can be substantial can thus be concentrated as point intake in a more rural area. Manufacturing industries have in developed countries often moved outside urban centres (where in the past they were often in city centres). Last but not least, mining activities often take place in remote areas, and large water abstractions can be very concentrated on a small rural spatial scale. When you produce a 0.5\*0.5 degree geo-dataset, these considerations can be very relevant. I acknowledge that the authors briefly describe limitations on page 12 lines 24-27. They also say this is a topic for further research. But please elaborate more on this, in the line with the argumentation I just made.*

Response: Thanks for your thoughtful comments. We agree with the referee that the spatial downscaling techniques for some sectors (e.g. electricity generation, mining and manufacturing) are rough. Water withdrawal for electricity generation is affected by many factors, including the location of power plants, the amount of generated electricity, generation type, cooling technology, and fuel types. As mentioned by the referee, water withdrawal for cooling can be concentrated outside urban centers for security reasons (e.g. nuclear power plants) and the need for large water quantities (e.g. along rivers). Also water withdrawal for mining and manufacturing are related to the geographic locations of mines and manufacturing centers, respectively. We have incorporated these points into the discussion of the limitations of the spatial downscaling techniques and future work in our revised manuscript.

*3) MISSING SECTOR TOURISM: The authors include 6 sectors, the ones which are typically identified for abstracting water. However, as in most studies, some particular water abstraction sectors are excluded. As indicated in the publication https://doi.org/10.1016/j.ecoser.2015.08.003, an important generally neglected sector is tourism. This includes water abstractions for snowmaking, which during winter months in mountain areas can be the largest regional water user (https://doi.org/10.2166/wst.2009.211 ). This water is generally taken from surface water, and is not accounted for in municipal water abstraction statistics. But this also includes water abstractions for hotels/swimming pools/spas both in winter and summer tourist areas (e.g. https://doi.org/10.1016/j.tourman.2013.05.010 ). These water users often have own private water abstractions, which are not accounted for in domestic/ municipal water use statistics. E.g. in Mediterranean regions during summer months these water abstractions can become shortly the dominant water use. Another touristic water user are golf courts (e.g. https://doi.org/10.1094/ATS-2009-0129-01-RS). These touristic water abstractions can on a local (0.5\*0.5 degree) and temporal (monthly) level be very significant. Please include in your discussion section a short subsection on this topic, based upon my input. Future research should include the sector tourism.*

Response: Thanks for your thoughtful input. We didn't consider tourism sector due to the lack of global water withdrawal dataset on tourism. In the revised manuscript, we have discussed the need for considering the missing sectors (e.g. tourism).

*4) SECTORS FORESTRY and AQUACULTURE: As indicated in the publication https://doi.org/10.1016/j.ecoser.2015.08.003, these sectors also account for water abstractions. Again, on a global level they may not be very significant in quantity, but on a local (0.5\*0.5 degree) and temporal (monthly) level, they can be very significant. Is forestry accounted for in your irrigation sector? Aquaculture can be very significant in a country like China. Please include in your discussion section a short subsection on this topic.*

Response: Thanks for your valuable inputs. Water use for forestry and aquaculture sector are important components of total water use. Here, aquaculture water withdrawal are included in livestock sector in our study, because FAO AQUASTAT provides water withdrawal for irrigation and total agricultural sector (i.e. water withdrawn for irrigation, livestock and aquaculture purposes), and livestock water withdrawal are calculated by the difference between agricultural and irrigation water withdrawal. We ignored water withdrawal for forestry sector in this study. In the revised manuscript, we have clarified that aquaculture water withdrawal is embedded in livestock water withdrawal, and have further discussed the significance of considering the forestry and aquaculture sectors in future work.

*5) DOMESTIC WATER ABSTRACTION: Please define in your paper what you mean with this. There is often confusion in the terminologies domestic and municipal water abstractions. There is a difference in water abstractions by households (generally defined as domestic water abstractions) and municipal water use, which additionally includes water use by shops, schools, public buildings ... and even for the cleaning of streets or public parks. As I understand your definition of "domestic sector" also includes these water users. Include a definition.*

Response: Thanks for your kind comments. Domestic water withdrawal in this study is the water use for indoor household purposes such as drinking, food preparation, bathing, washing clothes and dishes, flushing toilets, and outdoor purposes such as watering lawns and gardens, and also includes water use for the part of the industries and urban agriculture (e.g. water use by shops, schools, public buildings, and for the cleaning of streets or public parks). We have added the definition of water withdrawal by sectors in the revised version.

*MINOR COMMENTS*

*Page 2 line 19: You discussed the impact on the hydrological cycle and humans. Please add a sentence about the negative impact on the environment*

Response: Thanks for your kind comment, and we have revised the manuscript as suggested.

*page 2 Line 22 'We focus in this study on water withdrawal" - This is a choice, as also water consumption is an important statistic of water use. Water stress e.g. can be computed with both, as discussed in a recent publication https://doi.org/10.1016/j.scitotenv.2017.09.056*

Response: We agree. Water consumption is an important statistic of water use, and we also reconstructed the global gridded sectoral water consumption dataset, which will be also published together with water withdrawal data through an open-access link. Because the methods for reconstructing water consumption data are simple, we focus in this study on water withdrawal. The details of water consumption data have been represented in supplement materials.

*Page 3 line 7: Please add that also other selected European countries provide more detailed water use statistics (especially spatial data).*

Response: Thanks for your kind comment, and we have revised the manuscript as suggested.

*Page 3 Lines 18, 19: GHM and LSM - define abbreviation first*

Response: Thanks for your kind comment, and we have spelled out global hydrological model (GHM) and land surface model (LSM) for their first use in the revised manuscript.

*Page 4 Line 14 ... (GCAM): please add ref*

Response: Thanks for your kind comment, and we have added the references.

References:

Edmonds, J. et al., 1997. An integrated assessment of climate change and the accelerated introduction of advanced energy technologies-an application of MiniCAM 1.0. Mitigation and adaptation strategies for global change, 1(4): 311-339.

Kim, S.H., Edmonds, J., Lurz, J., Smith, S.J., Wise, M., 2006. The objECTS Framework for integrated Assessment: Hybrid Modeling of Transportation. The Energy Journal: 63-91.

*Page 6 Line 14: ... 30 urban centers ... : Urban water use characteristics can actually be quite different from rural water characteristics. By only downscaling based upon urban water use characteristics, the resulting dataset could be biased in temporal representation for more rural areas*

Response: We thank the reviewer for this excellent suggestion, and future work should certainly consider the distinction between rural and urban seasonal patterns. This will depend on the availability of monthly water use data in rural areas to facilitate such an exercise. As far as we know, there is no such a data product, and collecting monthly data for rural water is proved to be challenging as apparent by the number of countries with such data (Table 2). We have discussed this limitation in our revised manuscript.

*Page 8 Lines 17-21: Water abstraction for livestock: there are actually formulas that relate livestock water use to temperature.*

Response: Thanks for your comments. There are possible formulas that relate livestock water use to temperature. But we don't have monthly livestock water use data to parameterize such formulas. Thus, we applied the uniform distribution in this study. We have discussed about this point in the revised manuscript.

*Table 1: Please add a column with the spatial resolution of these datasets*

Response: Thanks for your comments. We have revised the table as suggested.

*Figure 3: (c) Electricity and not elecreicity.*

Response: Thanks for your comments. We have revised this in new manuscript.

---

## Author Comment (AC3) · 2 Feb 2018

**A point-by-point response to the reviews for "Reconstruction of global gridded monthly sectoral water withdrawals for 1971–2010 and analysis of their spatiotemporal patterns" by Zhongwei Huang et al**

Manuscript Details: Reconstruction of global gridded monthly sectoral water withdrawals for 1971-2010 and analysis of their spatiotemporal patterns, https://doi.org/10.5194/hess-2017-551

Authors: Zhongwei Huang, Mohamad Hejazi, Xinya Li, Qiuhong Tang, Chris Vernon, Guoyong Leng, Yaling Liu, Petra Döll, Stephanie Eisner, Dieter Gerten, Naota Hanasaki, Yoshihide Wada

We thank the reviewer for the very valuable comments and suggestions to improve the manuscript. Our point-by-point responses are listed below.

**Response to Anonymous Referee #3**

*Referee comments in Italics*

*This manuscript aims to reconstruct a global monthly gridded (0.5 degree) sectoral water withdrawal dataset for six water use sectors (irrigation, domestic, electricity generation (cooling of thermal power plants), livestock, mining, and manufacturing) for the period 1971-2010. And the reconstructed gridded water withdrawal dataset is open access. This paper is suitable for the HESS scope and also a valuable contribution to examining issues related to water withdrawals at fine spatial, temporal and sectoral scales.*

Response: We appreciate the positive and constructive feedback from the referee on our manuscript.

*The spatial distribution of water withdrawal for electricity generation depends on the distribution of the power plants. Most of the power plants are not concentrated in densely populated area. However, in this paper, spatial downscaling of water withdrawal for electricity generation (water withdrawal for cooling of thermal power plants) is based on population density maps. It should be future explained and discussed.*

Response: We agree that there are some limitations in the spatial downscaling of water withdrawal for electricity generation in this study, and future research should look into constructing a global database of power plants with details about their locations, construction year, fuel type, cooling technology, water source, generation capacity, capacity factor, etc. We have discussed this in details in the revised manuscript.

*In this paper, the spatial downscaling of water withdrawal for water withdrawal of electricity generation, domestic, mining and manufacturing was based on the population density maps. According to the gridded population map of the world (Center for International Earth Science Information Network (CIESIN) Columbia University), there are no people in Taklimakan Desert, some "no man's land" areas in Qinghai-Tibet Plateau, Sahara Desert. However, there are some water withdrawal of those sectors (please see Figure 5, 6, and 7). And in Figure S3, the dominant water withdrawal sector is manufacturing in Taklimakan Desert and some "no man's land" areas in Qinghai-Tibet Plateau, and is domestic in Sahara Desert. Please check it.*

Response: Thanks for your thoughtful comments. The gridded population maps we used for spatial downscaling are from HYDE during 1971-1989 and GPW during 1990-2010. Upon reassessing the two population products, we found that HYDE generally shows no population while GPW shows some population in these places (e.g., deserts and no-man lands). And since the results in Figure 5, 6 and 7 were all calculated based on the long-term annual mean value (1971-2010), the domestic, manufacturing, mining and electricity generations sectors which depend on population density will have water withdrawals in these "no-man" grids, but the withdrawals are very small. In the revised manuscript, grid cells with annual sectoral water withdrawal less than 0.01mm will be not taken into account when analyzing the spatiotemporal patterns, and related figures have been revised, and further discussion on the limitation in spatial downscaling techniques are also discussed.

---

## Author Response (AR1)

**A point-by-point response to the editor and reviewers for "Reconstruction of global gridded monthly sectoral water withdrawals for 1971–2010 and analysis of their spatiotemporal patterns" by Zhongwei Huang et al**

Manuscript Details: Reconstruction of global gridded monthly sectoral water withdrawals for 1971-2010 and analysis of their spatiotemporal patterns, https://doi.org/10.5194/hess-2017-551

Authors: Zhongwei Huang, Mohamad Hejazi, Xinya Li, Qiuhong Tang, Chris Vernon, Guoyong Leng, Yaling Liu, Petra Döll, Stephanie Eisner, Dieter Gerten, Naota Hanasaki, Yoshihide Wada

We thank the editor and the three reviewers for their very valuable comments and suggestions to improve the manuscript. Below we are our point-by-point responses.

**Response to Anonymous Referee #1**

*Referee comments in Italics*

*In this study, the authors reconstructed the global water withdrawal patterns from collected data by statistical downscaling. The spatial and temporal patterns of water withdrawal, along with sectoral divisions were analyzed. This work is not trivial. Estimating water withdrawal in a small watershed and considering various sectors is hard enough, not to say at the global scale. As a result, I do not think readers should blame the simplifications taken here.*

Response: We appreciate the positive and constructive feedback from the referee on our manuscript.

*However, I do have a concern of the irrigation part. It seems that the observations used for calibration is very sparse, especially in developing countries. For example, in the two major countries with water withdrawal – China and India, only data from West Bengal and Beijing were used. The result might be very biased because of the spatial variability of climate, water resources, and population density. Considering that 68% of water withdrawal is used for irrigation, this might lead to large errors in the final result.*

Response: We agree with the reviewer that irrigation is the largest water withdrawer and consumer of water globally and certainly in countries such as China and India. The mentioned example of using sparse data was specifically for the temporal downscaling of domestic water withdrawals, and not irrigation. For irrigation, we used the gridded irrigation water withdrawal estimates from Global Hydrological Models (GHMs) as a base layer to spatially and temporally downscale the reported country-level irrigation data from FAO AQUASTAT and USGS. As for the domestic sector, we collected monthly domestic water withdrawal from various sources (Table 2) to guide the temporal downscaling of domestic water withdrawals. As the referee mentioned, in the two major countries with water withdrawal – China and India, only data from West Bengal and Beijing were used. Given that domestic water withdrawal is roughly 7% of total water withdrawal in India and 12% in China, we acknowledge that more data would help improve the temporal downscaling of domestic water withdrawals, and future work should focus on collecting high resolution water withdrawal data both spatially and temporally. In the revised version of our manuscript, we discuss these aspects in detail.

*Also in Table 2: The second column is a mixture of cities, counties and states. In addition, it is better to indicate which state the city is located as it is not uncommon for multiple cities to have the same name.*

Response: Thanks for your kind comment. We have revised Table 2 as suggested.

*In Table 3, did you calibrate the R value in Japan and Spain too, or you just adopted the value from literature?*

Response: We just adopted the R value in Japan and Spain from literature because we don't have monthly water withdrawal data for these two countries. We clarify this distinction in the revised manuscript.

*Overall, the study is novel, the topic is suitable to HESS, and the manuscript is well written. I suggest a minor revision addressing my concerns mentioned above.*

Response: Thanks to referee for the positive feedback. We will revise our manuscript based on the suggestions and comments.

**Response to Anonymous Referee #2**

*Referee comments in Italics*

*GENERAL COMMENTS*

*This manuscript reconstructs gridded monthly water withdrawals globally for 6 sectors for 1971-2010 in a spatial resolution of 0.5\*0.5 degrees. The authors make this water*

*withdrawal dataset publicly available, which makes the manuscript more valuable and is in the line of the open-access philosophy of HESS.*

*Such a detailed global dataset is indeed to my knowledge the first in its kind and very useful. The statement at the end of the document (page 14 lines 3-5) "In whole, despite the uncertainties and limitations, this study is of great significance not only for cross comparison and validation for modeling and analyzing the impacts of human water use, but also for investigating water use related issues at finer spatial, temporal and sectoral scales" is very true.*

*I also appreciate that the authors include an extensive part in their manuscript on uncertainty (Section 4), as they acknowledge the uncertainty and limitations of their study.*

*The manuscript is novel, well written and in the scope of HESS. I recommend for moderate revision, as some issues need to be additionally addressed/discussed first.*

Response: We appreciate the positive and constructive feedback from the referee on our manuscript.

*MODERATE COMMENTS*

*1) The authors use as basis FAO AQUASTAT data and state-scale estimates of USGS for the US as basis for downscaling. Yet, on page 3 Lines 1-15 they argue that particular countries provide more detailed (especially spatially) data than the FAOSTAT data. This is indeed true for Germany as the authors point out, but also for many other European countries. These data (and additionally from Canada, China, ...) could have been used to optimise the downscaling methodology the authors use. Why was this choice made for the US but not for these other sources? I find this a bit a missed opportunity. I acknowledge that this means a lot more work, but you could have used all best data available instead of the US selection. Nevertheless, this does not have to be done within this paper, but maybe in future work. Please discuss shortly in the limitations section (section 4) of your manuscript.*

Response: Thanks for your thoughtful comments. We agree with the reviewer that we could have improved the spatial downscaling if we were to collect subnational sectoral water withdrawals based on the USGS equivalent agencies in these countries. We also agree with the reviewer that such an extension would amount for a lot of additional work and should be tackled in future research. Such an effort would also raise some additional challenges. For example, the definitions of sectoral water use are potentially inconsistent because these data are reported by various organizations and institutions. We only use

FAOSTAT and USGS data in this study, but we can update the open-access datasets when we get the subnational sectoral water withdrawal data in other regions or countries. In the revised manuscript, we further discuss the limitation and potential future work to improve the reconstructed dataset.

*2) SPATIAL DOWNSCALING TECHNIQUES: For some sectors (domestic, irrigation, livestock) the downscaling techniques are state of the art, for other sectors (electricity for cooling, mining and manufacturing) they are very rough. The three latter are based upon population-density maps. This is a very rough approach, as these sectors are in my opinion not always highly correlated with population densities. Water abstractions for cooling can very well be concentrated outside urban centres, for security reasons (e.g. nuclear power plants) and the availability of large water quantities (e.g. along rivers). Nulear water abstraction which can be substantial can thus be concentrated as point intake in a more rural area. Manufacturing industries have in developed countries often moved outside urban centres (where in the past they were often in city centres). Last but not least, mining activities often take place in remote areas, and large water abstractions can be very concentrated on a small rural spatial scale. When you produce a 0.5\*0.5 degree geo-dataset, these considerations can be very relevant. I acknowledge that the authors briefly describe limitations on page 12 lines 24-27. They also say this is a topic for further research. But please elaborate more on this, in the line with the argumentation I just made.*

Response: Thanks for your thoughtful comments. We agree with the referee that the spatial downscaling techniques for some sectors (e.g. electricity generation, mining and manufacturing) are rough. Water withdrawal for electricity generation are affected by many factors, including the location of power plants, the amount of generated electricity, generation type, cooling technology, and fuel types. As mentioned by referee, water withdrawal for cooling can be concentrated outside urban centres for security reasons (e.g. nuclear power plants) and the availability of large water quantities (e.g. along rivers), and water withdrawal for mining and manufacturing are also related to the geographic locations of manufacturing centers, and mines. In our revised manuscript, we elaborate more on the limitations and future work of the spatial downscaling techniques.

*3) MISSING SECTOR TOURISM: The authors include 6 sectors, the ones which are typically identified for abstracting water. However, as in most studies, some particular water abstraction sectors are excluded. As indicated in the publication https://doi.org/10.1016/j.ecoser.2015.08.003, an important generally neglected sector is tourism. This includes water abstractions for snowmaking, which during winter months in mountain areas can be the largest regional water user (https://doi.org/10.2166/wst.2009.211 ). This water is generally taken from surface water, and is not accounted for in municipal water abstraction statistics. But this also includes water abstractions for hotels/swimming pools/spas both in winter and summer tourist*

*areas (e.g. https://doi.org/10.1016/j.tourman.2013.05.010 ). These water users often have own private water abstractions, which are not accounted for in domestic/ municipal water use statistics. E.g. in Mediterranean regions during summer months these water abstractions can become shortly the dominant water use. Another touristic water user are golf courts (e.g. https://doi.org/10.1094/ATS-2009-0129-01-RS). These touristic water abstractions can on a local (0.5\*0.5 degree) and temporal (monthly) level be very significant. Please include in your discussion section a short subsection on this topic, based upon my input. Future research should include the sector tourism.*

Response: Thanks for your thoughtful input. We didn't consider tourism sector due to lack of global water withdrawal dataset on tourism. In the revised manuscript, we discuss the need for considering the missing sectors (e.g. tourism).

*4) SECTORS FORESTRY and AQUACULTURE: As indicated in the publication https://doi.org/10.1016/j.ecoser.2015.08.003, these sectors also account for water abstractions. Again, on a global level they may not be very significant in quantity, but on a local (0.5\*0.5 degree) and temporal (monthly) level, they can be very significant. Is forestry accounted for in your irrigation sector? Aquaculture can be very significant in a country like China. Please include in your discussion section a short subsection on this topic.*

Response: Thanks for your valuable inputs. Water use for forestry and aquaculture sector are important components of total water use. Here, aquaculture water withdrawal is included in livestock sector in our study, because FAO AQUASTAT provides water withdrawal for agricultural (i.e. water withdrawn for irrigation, livestock and aquaculture purposes) and livestock water withdrawal are calculated by the difference of agricultural and irrigation water withdrawal. We ignore water withdrawal for forestry sector in this study. In the revised manuscript, we will present the definition of water withdrawal by sectors, and further discuss the significance of considering the forestry and aquaculture sectors in our future work.

*5) DOMESTIC WATER ABSTRACTION: Please define in your paper what you mean with this. There is often confusion in the terminologies domestic and municipal water abstractions. There is a difference in water abstractions by households (generally defined as domestic water abstractions) and municipal water use, which additionally includes water use by shops, schools, public buildings ... and even for the cleaning of streets or public parks. As I understand your definition of "domestic sector" also includes these water users. Include a definition.*

Response: Thanks for your kind comments. Domestic water withdrawal in this study is the water use for indoor household purposes such as drinking, food preparation, bathing, washing clothes and dishes, flushing toilets, and outdoor purposes such as watering lawns

and gardens, and also includes water use for the part of the industries and urban agriculture (e.g. water use by shops, schools, public buildings, and for the cleaning of streets or public parks). We will add the definition of water withdrawal by sectors in the revised version.

*MINOR COMMENTS*

*Page 2 line 19: You discussed the impact on the hydrological cycle and humans. Please add a sentence about the negative impact on the environment*

Response: Thanks for your kind comment, and we will revise the manuscript as suggested.

*page 2 Line 22 'We focus in this study on water withdrawal" - This is a choice, as also water consumption is an important statistic of water use. Water stress e.g. can be computed with both, as discussed in a recent publication* *https://doi.org/10.1016/j.scitotenv.2017.09.056*

Response: We agree. Water consumption is an important statistic of water use, and we also reconstructed the global gridded sectoral water consumption dataset, which will be also published together with water withdrawal data through an open-access link. Because the methods for reconstructing water consumption data are simple, we focus in this study on water withdrawal. The details of water consumption data will be represented in supplement materials.

*Page 3 line 7: Please add that also other selected European countries provide more detailed water use statistics (especially spatial data).*

Response: Thanks for your kind comment, and we will revise the manuscript as suggested.

*Page 3 Lines 18, 19: GHM and LSM - define abbreviation first*

Response: Thanks for your kind comment, and we will define abbreviation first in revised manuscript.

*Page 4 Line 14 ... (GCAM): please add ref*

Response: Thanks for your kind comment, and we will add the references.

*Page 6 Line 14: ... 30 urban centers ... : Urban water use characteristics can actually be quite different from rural water characteristics. By only downscaling based upon urban*

*water use characteristics, the resulting dataset could be biased in temporal representation for more rural areas*

Response: That is an excellent suggestion, and future work should certainly capture such a distinction between rural and urban seasonal patterns. This obviously will hinge on the availability of such data to facilitate such an exercise. As far as we know, there is not such a product, and collecting monthly data for this sector proved to be challenging as apparent by the number of countries with such data (Table 2). We will discuss this limitation in our revised manuscript.

*Page 8 Lines 17-21: Water abstraction for livestock: there are actually formulas that relate livestock water use to temperature.*

Response: Thanks for your comments. There are possible formulas that relate livestock water use to temperature. But we don't have monthly livestock water use data to parameterize such formulas. Thus, we use the uniform distribution in this study.

*Table 1: Please add a column with the spatial resolution of these datasets*

Response: Thanks for your comments. We will revise the table as suggested.

*Figure 3: (c) Electricity and not elecreicity.*

Response: Thanks for your comments. We will revise this in new manuscript.

**Response to Anonymous Referee #3**

*Referee comments in Italics*

*This manuscript aims to reconstruct a global monthly gridded (0.5 degree) sectoral water withdrawal dataset for six water use sectors (irrigation, domestic, electricity generation (cooling of thermal power plants), livestock, mining, and manufacturing) for the period 1971-2010. And the reconstructed gridded water withdrawal dataset is open access. This paper is suitable for the HESS scope and also a valuable contribution to examining issues related to water withdrawals at fine spatial, temporal and sectoral scales.*

Response: We appreciate the positive and constructive feedback from the referee on our manuscript.

*The spatial distribution of water withdrawal for electricity generation depends on the distribution of the power plants. Most of the power plants are not concentrated in densely populated area. However, in this paper, spatial downscaling of water withdrawal for*

*electricity generation (water withdrawal for cooling of thermal power plants) is based on population density maps. It should be future explained and discussed.*

Response: We agree that there are some limitations in the spatial downscaling of water withdrawal for electricity generation, and future research should look into constructing a global database of power plants with details about their locations, construction year, fuel type, cooling technology, water source, generation capacity, capacity factor, etc. We discuss this in details in the revised manuscript.

*In this paper, the spatial downscaling of water withdrawal for water withdrawal of electricity generation, domestic, mining and manufacturing was based on the population density maps. According to the gridded population map of the world (Center for International Earth Science Information Network (CIESIN) Columbia University), there are no people in Taklimakan Desert, some "no man's land" areas in Qinghai-Tibet Plateau, Sahara Desert. However, there are some water withdrawal of those sectors (please see Figure 5, 6, and 7). And in Figure S3, the dominant water withdrawal sector is manufacturing in Taklimakan Desert and some "no man's land" areas in Qinghai-Tibet Plateau, and is domestic in Sahara Desert. Please check it.*

Response: Thanks for your thoughtful comments. The gridded population maps we used for spatial downscaling are from HYDE during 1971-1989 and GPW during 1990-2010. Upon reassessing the two population products, we found that HYDE generally shows no population while GPW shows some population in these places (e.g., deserts and no-man lands). And since the results in Figure 5, 6 and 7 were all calculated based on the long-term annual mean value (1971-2010), the domestic, manufacturing, mining and electricity generations sectors which depend on population density will have water withdrawals in these "no-man" grids, but the withdrawals are very small. In the revised manuscript, we will revise the figures with consideration of "no man's land" area based on your suggestions, and further discuss the limitation in spatial downscaling techniques.

**A list of all relevant corrections made in the manuscript for "Reconstruction of global gridded monthly sectoral water withdrawals for 1971–2010 and analysis of their spatiotemporal patterns" by Zhongwei Huang et al**

Manuscript Details: Reconstruction of global gridded monthly sectoral water withdrawals for 1971-2010 and analysis of their spatiotemporal patterns, https://doi.org/10.5194/hess-2017-551

Authors: Zhongwei Huang, Mohamad Hejazi, Xinya Li, Qiuhong Tang, Chris Vernon, Guoyong Leng, Yaling Liu, Petra Döll, Stephanie Eisner, Dieter Gerten, Naota Hanasaki, Yoshihide Wada

We thank three reviewers and editor for their very valuable comments. Below is a list of all relevant changes made in the manuscript.

Relevant changes made in the manuscript are as follows:

1) A co-author was added due to his great contribution to data publication and the manuscript revision.
2) The reconstruction dataset are available online, and a link was added in the manuscript.
3) More details about the definition of the sectoral water withdrawal were added in the manuscript.
4) Limitations and future works in data source and the methods we used were further discussed.
5) Some figures were corrected by ignoring area with annual sectoral water withdrawal less than 0.01mm when analysis the temporal pattern and changing trend of sectoral water withdrawal.
6) In the supplement material, methods used for generating sectoral water consumption data are represented; and a new figure about the changes in global water withdrawal by 6 sectors during 1971-2010 was added.

The following pages are a marked-up manuscript version and supplementary. We hope that the revisions in the manuscript and our accompanying responses will be sufficient to make our manuscript suitable for publication in HESS.

[revised manuscript text omitted]

1. Methods used for generating sectoral water consumption data

Besides the global gridded monthly sectoral water withdrawal data during 1971-2010, sectoral water consumption data were also produced at grid scale and monthly time step and are available online (https://doi.org/10.5281/zenodo.897933). For the irrigation sector, irrigation water consumption was generated by reapplying the correction factor in Eq.(1) to irrigation water consumptions simulated by four GHMs (i.e. WaterGAP, LPJmL , H08, and PCR-GLOBWB):

$$Cir_{i,j,g} = Cir\_sim_{i,j,g} \times f_{m,p} ; \qquad (S1)$$

Where $Cir_{i,j,g}$ is the reconstructed irrigation water consumption for the month $i$ of year $j$ at grid $g$ (m3), and $Cir\_sim_{i,j,g}$ is the irrigation water consumption for the month $i$ of year $j$ at grid $g$ simulated by four GHMs (m3); $f_{m,p}$ is the correction factor calculated in Eq.(1). For the remaining sectors, consumptive water use efficiency (the proportion of water consumption to water withdrawal) was used. Based on the simulation of Flörke et al (2013), consumptive water use efficiencies for electricity generation, domestic and manufacturing sector were calculated at country level, and global consumptive water use efficiencies for livestock and mining adopted the value in the US which was estimated by USGS. Thus, water consumptions by these 5 sectors were calculated by the products of reconstructed water withdrawal data and the consumptive water use efficiencies.

**2. Supplement figures**

Figure S1. Simulated annual irrigation water withdrawal using each of the following four GHMs (i.e., WaterGAP, H08, LPJmL, and PCR-GLOBWB) in comparison with FAO AQUASTAT data at country level and USGS estimation at state level.

[Figure]

Figure S2. Mean monthly irrigation water withdrawal (normalized in percentage) in 32 GCAM regions simulated by four GHMs (i.e., WaterGAP, H08, LPJmL, and PCR-GLOBWB).

[Figure]

Figure S3. Spatial distribution of global dominant water withdrawal sectors.

[Figure]

Figure S4. Water withdrawal by 6 sectors during 1971-2010 in (a) Global, (b) China, (c) the US, (d) India and (e) EU27.

[Figure]

Figure  S5 Monthly and annual time-series of total water withdrawal for global, China, the US, India and EU27 during 1971-2010

[Figure]

Figure S6. Monthly and annual time-series of irrigation water withdrawal for global, China, US, India and EU27 during 1971-2010.

[Figure]

Figure S6S7. Monthly and annual time-series of domestic water withdrawal for global, China, US, India and EU27 during 1971-2010.

[Figure]

Figure  S8 Monthly and annual time-series of electricity generation water withdrawal for global, China, US, India and EU27 during 1971-2010

[Figure]

Figure S9. Annual time-series of water withdrawal by sector (mining, livestock, and manufacturing) for global, China, US, India and EU27 during 1971-2010.

[Figure]

Figure  S10 comparison of global water withdrawal used in this study with estimates from Flörke et al. (2013) and Shiklomanov (2000) for domestic and industrial sectors.

[Figure]

Figure  S11 coefficient of variation (CV) of irrigation water withdrawal in JJA and DJF caused by multi-model framework and by multi-forcing data: December to February (DJF) and June to August (JJA).

[Figure]

Figure S11 S12 coefficient of variation (CV)  caused by different climate forcing in temporal downscaling of (a) domestic and (b) electricity generation in 4 seasons: : December to February (DJF), March to May (MAM), June to August (JJA) and September to November (SON).

[Figure]